# Rethinking Model-based, Policy-based, and Value-based Reinforcement Learning via the Lens of Representation Complexity

**Guhao Feng**[*†]        **Han Zhong**[*‡]

## Abstract

Reinforcement Learning (RL) encompasses diverse paradigms, including model-based RL, policy-based RL, and value-based RL, each tailored to approximate the model, optimal policy, and optimal value function, respectively. This work investigates the potential hierarchy of representation complexity among these RL paradigms. By utilizing computational complexity measures, including time complexity and circuit complexity, we theoretically unveil a potential representation complexity hierarchy within RL. We find that representing the model emerges as the easiest task, followed by the optimal policy, while representing the optimal value function presents the most intricate challenge. Additionally, we reaffirm this hierarchy from the perspective of the expressiveness of Multi-Layer Perceptrons (MLPs), which align more closely with practical deep RL and contribute to a completely new perspective in theoretical studying representation complexity in RL. Finally, we conduct deep RL experiments to validate our theoretical findings.

## 1 Introduction

The past few years have witnessed the tremendous success of Reinforcement Learning (RL) [43] in solving intricate real-world decision-making problems, such as Go [41] and robotics [27]. These successes can be largely attributed to powerful function approximators, particularly Neural Networks (NN) [30], and the evolution of modern RL algorithms. These algorithms can be categorized into *model-based RL*, *policy-based RL*, and *value-based RL* based on their respective objectives of approximating the underlying model, optimal policy, or optimal value function.

Despite the extensive theoretical analysis of RL algorithms in terms of statistical error [e.g., 5, 21, 42, 24, 23, 14, 17, 58, 51] and optimization error [e.g., 2, 49, 9, 29] lenses, a pivotal perspective often left in the shadows is *approximation error*. Specifically, the existing literature predominantly relies on the (approximate) realizability assumption, assuming that the given function class can sufficiently capture the underlying model, optimal value function, or optimal policy. However, limited works examine the *representation complexity* in different RL paradigms — the complexity of the function class needed to represent the underlying model, optimal policy, or optimal value function. In particular, the following problem remains elusive:

*Is there a representation complexity hierarchy among model-based RL, policy-based RL, and value-based RL?*

To our best knowledge, the theoretical exploration of this question is limited, with only two exceptions [13, 62]. [13] employs piecewise linear functions to represent both the model and value functions,

---

[*]Alphabetical order.
[†]School of EECS, Peking University. Email: `fenguhao@stu.pku.edu.cn`
[‡]Center for Data Science, Peking University. Email: `hanzhong@stu.pku.edu.cn`

38th Conference on Neural Information Processing Systems (NeurIPS 2024).

| | | Computational Complexity (time complexity and circuit complexity) | Expressiveness of Log-precision MLP/Transformer (constant layers and polynomial hidden dimension) |
|---|---|---|---|
| 3-SAT MDP NP MDP | Model | $AC^0$ | ✓ |
| | Policy | NP-Complete | ✗ |
| | Value | NP-Complete | ✗ |
| CVP MDP P MDP | Model | $AC^0$ | ✓ |
| | Policy | $AC^0$ | ✓ |
| | Value | P-Complete | ✗ |

Table 1: A summary of our main results. ✓ means that the function can be represented by log-precision MLP with constant layers and polynomial hidden dimension, while ✗ means that this MLP class cannot represent the function. Blue denotes low representation complexity, and red represents high representation complexity.

utilizing the number of linear pieces as a metric for representation complexity. They construct a class of Markov Decision Processes (MDPs) where the underlying model can be represented by a constant piecewise linear function, while the optimal value function necessitates an exponential number of linear pieces for representation. This disparity underscores that the model's representation complexity is comparatively less than that of value functions. Recently, [62] reinforced this insight through a more rigorous circuit complexity perspective. They introduce a class of MDPs wherein the model can be represented by circuits with polynomial size and constant depth, while the optimal value function cannot. However, the separation between model-based RL and value-based RL demonstrated in [62] may not be deemed significant (cf. Remark 5.6). More importantly, [13, 62] do not consider policy-based RL and do not connect the representation complexity in RL with the expressive power of neural networks such as Multi-Layer Perceptron (MLP) and Transformers [47], thereby providing limited insights for deep RL.

**Our Contributions.** To address the limitations of previous works and provide a comprehensive understanding of representation complexity in RL, we explore representation complexity within **(1)** model-based RL; **(2)** policy-based RL; and **(3)** value-based RL, employing **(i)** computational complexity (time complexity and circuit complexity) and **(ii)** expressiveness of MLP to rigorously quantify representation complexity. We outline our results below, further summarized in Table 1.

- To elucidate the representation complexity separation between model-based RL and model-free RL[4], we introduce two types of MDPs: 3-SAT MDPs (Definition 3.2) and a broader class referred to as NP MDPs (Definition 3.7). In both cases, the representation complexity of the model, inclusive of the reward function and transition kernel, falls within the complexity class $AC^0$ (cf. Section 2). In contrast, we demonstrate that the representation of the optimal policy and optimal value function for 3-SAT MDPs and NP MDPs is NP-complete. Significantly, our results not only demonstrate the significant representation complexity separation between model-based RL and model-free RL, but also address an open question in [62]. See Remark 3.4 for details.

- To further showcase the separation within the realm of model-free RL, we introduce two classes of MDPs: CVP MDPs (Definition 4.1) and a broader category denoted as P MDPs (Definition 4.4). In both instances, the representation complexity of the underlying model and the optimal policy is confined to the complexity class $AC^0$. In contrast, the representation complexity for the optimal value function is characterized as P-complete, reflecting the inherent computational challenges associated with computing optimal values. This underscores the efficiency in representing policies (and models) while emphasizing the inherent representation complexity involved in determining optimal value functions.

- To provide more practical insights, we establish a connection between our previous findings and the realm of deep RL, where the model, policy, and value function are represented by neural networks. Specifically, for 3-SAT MDPs and NP MDPs, we demonstrate the effective representation of the model through a constant-layer MLP/Transformer with polynomial hidden dimension, while the optimal policy and optimal value exhibit constraints in such representation. Furthermore, for the CVP MDPs and P MDPs, we illustrate that both the underlying model and optimal policy can be represented by constant-layer MLPs/Transformers with polynomial hidden dimension. However, the optimal value, in contrast, faces limitations in its representation using MLPs/Transformers with constant layers and polynomial hidden dimensions. These results corroborate the messages conveyed through the perspective of computational complexity, contributing a novel perspective that bridges the representation complexity in RL with the expressive power of MLP/Transformer. In addition, we conduct deep RL experiments to corroborate our theory.

---

[4]Throughout this paper, we use the term model-free RL to include both policy-based RL and value-based RL.

In summary, our work unveils a potential hierarchy in representation complexity among different categories of RL paradigms — where the underlying model is the most straightforward to represent, followed by the optimal policy, and the optimal value function emerges as the most intricate to represent. This insight offers valuable guidance on determining appropriate targets for approximation, enhancing understanding of the inherent challenges in representing key elements across different RL paradigms. Moreover, our representation complexity theory is closely tied to the sample efficiency gap observed among various RL paradigms. Given that the sample complexity of RL approaches often depends on the realizable function class in use [21, 42, 14, 23, 17, 58], our results suggest that representation complexity may play a significant role in determining the diverse sample efficiency achieved by different RL algorithms. This aligns with the observed phenomenon that model-based RL typically exhibits superior sample efficiency compared to other paradigms [22, 42, 45, 20, 54, 56]. Consequently, our work underscores the importance of considering representation complexity in the design of sample-efficient RL algorithms. See Appendix C.1 for more elaborations.

**Additional Related Works.** We have discussed most related works in the introduction part, and more related works are deferred to Appendix A.

**Notations.** We denote the distribution over a set $\mathcal{X}$ by $\Delta(\mathcal{X})$. We use $\mathbb{N}$ and $\mathbb{N}_+$ to denote the set of all natural numbers and positive integers, respectively. For any $n \in \mathbb{N}_+$, we denote $[n] = \{1, 2, \cdots, n\}$, and $\mathbf{0}_n = \underbrace{(0, 0, \cdots, 0)}_{n \text{ times}}^\top, \mathbf{1}_n = \underbrace{(1, 1, \cdots, 1)}_{n \text{ times}}^\top$.

## 2 Preliminaries

**Markov Decision Process.** We consider the finite-horizon Markov decision process (MDP), which is defined by a tuple $\mathcal{M} = (\mathcal{S}, \mathcal{A}, H, \mathcal{P}, r)$. Here $\mathcal{S}$ is the state space, $\mathcal{A}$ is the action space, $H$ is the length of each episode, $\mathcal{P} : \mathcal{S} \times \mathcal{A} \mapsto \Delta(\mathcal{S})$ is the transition kernel, and $r : \mathcal{S} \times \mathcal{A} \mapsto [0, 1]$ is the reward function. Moreover, when the transition kernel is deterministic, say $\mathcal{P}(\cdot \mid s, a) = \delta_{s'}$ for some $s' \in \mathcal{S}$. we denote $\mathcal{P}(s, a) = s'$. A policy $\pi = \{\pi_h : \mathcal{S} \mapsto \Delta(\mathcal{A})\}_{h=1}^H$ consists of $H$ mappings from the state space to the distribution over action space. For the deterministic policy $\pi_h$ satisfying $\pi_h(\cdot|s) = \delta_a$ for some $a \in \mathcal{A}$, we denote $\pi_h(s) = a$. Here $\delta_a$ is the Dirac measure at $a$. Given a policy $\pi$, for any $(s, a) \in \mathcal{S} \times \mathcal{A}$, we define the state value function and state-action value function (Q-function) as $V_h^\pi(s) = \mathbb{E}_\pi[\sum_{h'=h}^H r_{h'}(s_{h'}, a_{h'}) \mid s_h = s], Q_h^\pi(s, a) = \mathbb{E}_\pi[\sum_{h'=h}^H r_{h'}(s_{h'}, a_{h'}) \mid s_h = s, a_h = a]$. Here the expectation $\mathbb{E}_\pi[\cdot]$ is taken with respect to the randomness incurred by the policy $\pi$ and transition kernels. There exists an optimal policy $\pi^*$ achieves the highest value at all timesteps and states, i.e., $V_h^{\pi^*}(s) = \sup_\pi V_h^\pi(s)$ for any $h \in [H]$ and $s \in \mathcal{S}$. For notation simplicity, we use the shorthands $V_h^* = V_h^{\pi^*}$ and $Q_h^* = Q_h^{\pi^*}$ for any $h \in [H]$.

RL encompasses diverse paradigms, including model-based RL, policy-based RL, and value-based RL, each tailored to approximate the model ($r$ and $\mathcal{P}$), optimal policy $\pi^*$, and optimal value function $Q^*$, respectively. See Appendix B.1 for more details regarding function approximation.

**Computational Complexity.** Our work will use some classical computational complexity theory [3]. Specifically, we will utilize five complexity classes: $\mathsf{AC}^0$, $\mathsf{TC}^0$, $\mathsf{L}$, $\mathsf{P}$, and $\mathsf{NP}$. Here, $\mathsf{L}$, $\mathsf{P}$, and $\mathsf{NP}$ are defined in terms of the running time of Turing Machines (Definition B.1), while $\mathsf{AC}^0$ and $\mathsf{TC}^0$ are defined based on the complexity of Boolean circuits (Definition B.2). To facilitate the readers, we provide the detailed definition in Appendix B.2. The relationship between these five complexity classes is $\mathsf{AC}^0 \subsetneq \mathsf{TC}^0 \subset \mathsf{L} \subset \mathsf{P} \subset \mathsf{NP}$. The question of whether the relationship $\mathsf{TC}^0 \subset \mathsf{P} \subset \mathsf{NP}$ holds as a strict inclusion remains elusive in theoretical computer science. However, it is widely conjectured that $\mathsf{P} = \mathsf{NP}$ and $\mathsf{TC}^0 = \mathsf{P}$ are unlikely to be true.

## 3 The Separation between Model-based RL and Model-free RL

In this section, we focus on the representation complexity gap between model-based RL and model-free RL, which encompasses both policy-based RL and value-based RL.

### 3.1 3-SAT MDP

As a warmup example to showcase the separation between model-based RL and model-free RL, we propose the 3-SAT MDPs, whose construction is closely linked to the well-known NP-complete problem 3-satisfiability (3-SAT). The formal definition of 3-SAT is as follows.

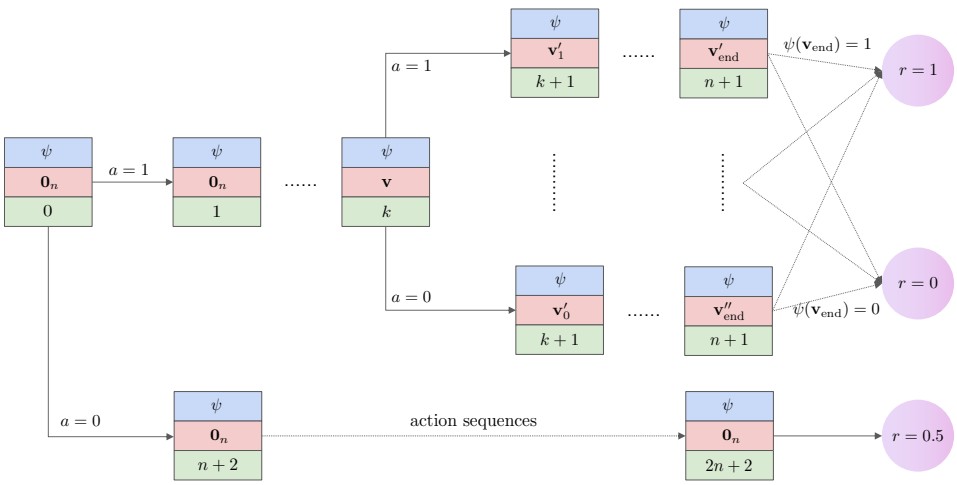

Figure 1: A visualization of 3-SAT MDPs. Here, $\mathbf{v}$ is an $n$-dimensional vector, $\mathbf{v}_0$ and $\mathbf{v}_1$ are vectors obtained by replacing the $k$-th element of $\mathbf{v}$ with 0 and 1, respectively. Additionally, $\mathbf{v}_{\text{end}}$, $\mathbf{v}'_{\text{end}}$, and $\mathbf{v}''_{\text{end}}$ represent the assignments at the end of the episode.

**Definition 3.1** (3-SAT Problem). A Boolean formula $\psi$ over variables $u_1, u_2, \cdots, u_n$ is in the 3-conjunctive normal form (3-CNF) if it takes the form of a conjunction of one or more disjunctions, each containing exactly 3 literals. Here, a literal refers to either a variable or the negation of a variable. Formally, the 3-CNF formula $\psi$ has the form of $\psi = \bigwedge_{i \in \mathcal{I}} (v_{i,1} \vee v_{i,2} \vee v_{i,3})$, where $\mathcal{I}$ is the index set and $v_{i,j} \in \{u_k, \neg u_k\}$ for some $k \in [n]$. The 3-SAT problem is defined as follows: given a 3-CNF Boolean formula $\psi$, judge whether $\psi$ is satisfiable. Here, "satisfiable" means that there exists an assignment of variables such that the formula $\psi$ evaluates to 1.

Now we present the detailed construction of 3-SAT MDPs.

**Definition 3.2** (3-SAT MDP). For any $n \in \mathbb{N}_+$, let $\mathcal{V} = \{u_1, \neg u_1, \cdots, u_n, \neg u_n\}$ be the set of literals. An $n$-dimensional 3-SAT MDP $(\mathcal{S}, \mathcal{A}, H, \mathcal{P}, r)$ is defined as follows. The state space $\mathcal{S}$ is defined by $\mathcal{S} = \mathcal{V}^{3n} \times \{0,1\}^n \times (\{0\} \cup [2n+2])$, where each state $s$ can be denoted as $s = (\psi, \mathbf{v}, k)$. In this representation, $\psi$ is a 3-CNF formula consisting of $n$ clauses and represented by its $3n$ literals, $\mathbf{v} \in \{0,1\}^n$ can be viewed as an assignment of the $n$ variables and $k$ is an integer recording the number of actions performed. The action space is $\mathcal{A} = \{0,1\}$ and the planning horizon is $H = n + 2$. Given a state $s = (\psi, \mathbf{v}, k)$, for any $a \in \mathcal{A}$, the reward $r(s,a)$ is defined by:

$$r(s,a) = \begin{cases} 1 & \text{If } \mathbf{v} \text{ is satisfiable and } k = n+1, \\ 0.5 & \text{If } k = 2n+2, \\ 0 & \text{Otherwise.} \end{cases} \tag{3.1}$$

Moreover, the transition kernel is deterministic and takes the following form:

$$\mathcal{P}\big((\psi, \mathbf{v}, k), a\big) = \begin{cases} (\psi, \mathbf{v}, n+2) & \text{If } a = k = 0, \\ (\psi, \mathbf{v}, 1) & \text{If } a = 1 \text{ and } k = 0, \\ (\psi, \mathbf{v}', k+1) & \text{If } k \in [n] \\ (\psi, \mathbf{v}, k+1) & \text{If } k > n. \end{cases} \tag{3.2}$$

where $\mathbf{v}'$ is obtained from $\mathbf{v}$ by setting the $k$-th bit as $a$ and leaving other bits unchanged, i.e., $\mathbf{v}'[k] = a$ and $\mathbf{v}'[k'] = \mathbf{v}[k']$ for $k' \neq k$. The initial state takes form $(\psi, \mathbf{0}_n, 0)$ for any length-$n$ 3-CNF formula $\psi$.

The visualization of 3-SAT MDPs is given in Figure 1. We assert that our proposed 3-SAT model is relevant to real-world problems. In the state $s = (\psi, \mathbf{v}, k)$, $\psi$ characterizes intrinsic environmental factors that remain unchanged by the agent, while $\mathbf{v}$ and $k$ represent elements subject to the agent's influence. Notably, the agent is capable of changing $\mathbf{0}_n$ to any $n$-bits binary string within the episode. Using autonomous driving as an example, $\psi$ could denote fixed factors like road conditions and weather, while $\mathbf{v}$ and $k$ may represent aspects of the car that the agent can control. While states and actions in practical scenarios might be continuous, they are eventually converted to binary strings in computer storage due to bounded precision. Regarding the reward structure, the agent only receives rewards at the end of the episode, reflecting the goal-conditioned RL setting and the sparse reward

setting, which capture many real-world problems. Intuitively, the agent earns a reward of $1$ if $\psi$ is satisfiable, and the agent transforms $\mathbf{0}_n$ into a variable assignment that makes $\psi$ equal to $1$ through a sequence of actions. The agent receives a reward of $0.5$ if, at the first step, it determines that $\psi$ is unsatisfiable and chooses to "give up". Here, we refer to taking action $0$ at the first step as "give up" since this action at the outset signifies that the agent foregoes the opportunity to achieve the highest reward of $1$. In all other cases, the agent receives a reward of $0$. Using the example of autonomous driving, if the car successfully reaches its (reachable) destination, it obtains the highest reward. If the destination is deemed unreachable and the car chooses to give up at the outset, it receives a medium reward. This decision is considered a better choice than investing significant resources in attempting to reach an unattainable destination, which would result in the lowest reward.

**Theorem 3.3** (Representation complexity of 3-SAT MDP). *Let $\mathcal{M}_n$ be the $n$-dimensional 3-SAT MDP in Definition 3.2. The transition kernel $\mathcal{P}$ and the reward function $r$ of $\mathcal{M}_n$ can be computed by circuits with polynomial size (in $n$) and constant depth, falling within the circuit complexity class $\mathsf{AC}^0$. However, computing the optimal value function $Q_1^*$ and the optimal policy $\pi_1^*$ of $\mathcal{M}_n$ are both NP-complete under the polynomial time reduction.*

The proof of Theorem 3.3 is deferred to Appendix E.1. Theorem 3.3 states that the representation complexity of the underlying model is in $\mathsf{AC}^0$, whereas the representation complexity of optimal value function and optimal policy is NP-complete. This demonstrates the significant separation of the representation complexity of model-based RL and that of model-free RL.

**Remark 3.4.** The recent work of [62] raises an open problem regarding the existence of a class of MDPs whose underlying model can be represented by $\mathsf{AC}^k$ circuits while the optimal value function cannot. Here, $\mathsf{AC}^k$ is a complexity class satisfying $\mathsf{AC}^0 \subset \mathsf{AC}^k \subset \mathsf{P} \subset \mathsf{NP}$. Therefore, our results not only address this open problem by revealing a more substantial gap in representation complexity between model-based RL and model-free RL but also surpass the expected resolution conjectured in [62].

**Remark 3.5.** Although Theorem 3.3 only shows that $Q_1^*$ is hard to represent, our proof also implies that $V_1^*$ is hard to represent. Moreover, we can extend our results to the more general case, say $\{Q_h^*\}_{h \in [\lfloor \frac{n}{2} \rfloor]}$ are NP-complete, by introducing additional irrelevant steps. Notably, one cannot anticipate $Q_h^*$ to be hard to represent for any $h \in [H]$ since $Q_H$ reduces to the one-step reward function $r$. This aligns with our intuition that the multi-step correlation is pivotal in rendering the optimal value functions in the "early steps" challenge to represent. Also, although we only show that $\pi_1^*$ is hard to represent in our proof, the result can be extended to step $h$ where $2 \le h \le H$, as $\pi_1^*$ also serves as an optimal policy at step $h$. Finally, We want to emphasize that, since our objective is to show that $Q^* = \{Q_h^*\}_{h=1}^H$ and $\pi^* = \{\pi_h^*\}_{h=1}^H$ have high representation complexity, it suffices to demonstrate that $Q_1^*$ and $\pi_1^*$ are hard to represent.

**Remark 3.6.** Our results can be extended to stochastic MDPs and partially observable MDPs (POMDPs) via slight modifications. See Appendices H.1 and H.2 for details.

## 3.2 NP **MDP: A Broader Class of MDPs**

We extend the results for 3-SAT MDPs by introducing the concept of NP MDP—a broader class of MDPs. Specifically, for any NP-complete language $\mathcal{L}$, we can construct a corresponding NP MDP that encodes $\mathcal{L}$ into the structure of MDPs. Importantly, this broader class of MDPs yields the same outcomes as 3-SAT MDPs. In other words, in the context of NP MDP, the underlying model can be computed by circuits in $\mathsf{AC}^0$, while the computation of both the optimal value function and optimal policy remains NP-complete. The detailed definition of NP MDP is provided below.

**Definition 3.7** (NP MDP). An NP MDP is defined concerning a language $\mathcal{L} \in \mathsf{NP}$. Let $M$ be a nondeterministic Turing Machine recognizing $\mathcal{L}$ in at most $P(n)$ steps, where $n$ is the length of the input string and $P(n)$ is a polynomial. Let $M$ have valid configurations denoted by $\mathcal{C}_n$, and let each configuration $c = (s_M, \mathbf{t}, l) \in \mathcal{C}_n$ encompass the state of the Turing Machine $s_M$, the contents of the tape $\mathbf{t}$, and the pointer on the tape $l$, requiring $O(P(n))$ bits for representation. Then an $n$-dimensional NP MDP is defined as follows. The state space $\mathcal{S}$ is $\mathcal{C}_n \times (\{0\} \cup [2P(n) + 2])$, and each $s = (c, k) \in \mathcal{S}$ consists of a valid configuration $c = (s_M, \mathbf{t}, l)$ and a index $k \in \{0\} \cup [2P(n) + 2]$ recording the number of steps $M$ has executed. The action space is $\mathcal{A} = \{0, 1\}$ and the planning horizon is $H = P(n) + 2$. Given state $s = (c, k) = (s_M, \mathbf{t}, l, k)$ and action $a$, the reward function $r(s, a)$ is defined by:

$$r(s, a) = \begin{cases} 1 & \text{If } s_M = s_{\text{accpet}} \text{ and } k = P(n) + 1, \\ 0.5 & \text{If } k = 2P(n) + 2, \\ 0 & \text{Otherwise,} \end{cases} \tag{3.3}$$

where $s_{\text{accept}}$ is the accept state of Turing Machine $M$. Moreover, the transition kernel is deterministic and can be defined as follows:

$$\mathcal{P}\big((c,k),a\big) = \begin{cases} (c, P(n)+2) & \text{If } a = k = 0, \\ (c, 1) & \text{If } a = 1 \text{ and } k = 0, \\ (c', k+1) & \text{If } k \in [P(n)] \\ (c, k+1) & \text{If } k > P(n). \end{cases} \tag{3.4}$$

where $c'$ is the configuration obtained from $c$ by selecting the branch $a$ at the current step and executing the Turing Machine $M$ for one step. Let $\mathbf{x}_{\text{input}}$ be an input string of length $n$. We can construct the initial configuration $c_0$ of the Turing Machine $M$ on the input $\mathbf{x}_{\text{input}}$ by copying the input string onto the tape, setting the pointer to the initial location, and designating the state of the Turing Machine as the initial state. The initial state is defined as $(c_0, 0)$.

The definition of NP MDP in Definition 3.7 generalizes that of the 3-SAT MDP in Definition 3.2 by incorporating the nondeterministic Turing Machine, a fundamental computational mode. The configuration $c$ and the accept state $s_{\text{accpet}}$ in NP MDPs mirror the formula-assignment pair $(\psi, \mathbf{v})$ and the scenario that $\psi(\mathbf{v}) = 1$ in 3-SAT MDP, respectively. To the best of our knowledge, NP MDP is the first class of MDPs defined in the context of (non-deterministic) Turing Machine and can encode *any* NP-complete problem in an MDP structure. This represents a significant advancement compared to the Majority MDP in [62] and the 3-SAT MDP in Definition 3.2, both of which rely on specific computational problems such as the Majority function and the 3-SAT problem. The following theorem provides the theoretical guarantee for the NP-complete MDP.

**Theorem 3.8** (Representation complexity of NP MDP). Consider any NP-complete language $\mathcal{L}$ alongside its corresponding $n$-dimensional NP MDP $\mathcal{M}_n$, as defined in Definition 3.7. The transition kernel $\mathcal{P}$ and the reward function $r$ of $\mathcal{M}_n$ can be computed by circuits with polynomial size (in $n$) and constant depth, belonging to the complexity class $\mathsf{AC}^0$. In contrast, the problems of computing the optimal value function $Q_1^*$ and the optimal policy $\pi_1^*$ of $\mathcal{M}_n$ are both NP-complete under the polynomial time reduction.

The proof of Theorem 3.8 is deferred to Appendix E.2. Theorem 3.8 demonstrates that a substantial representation complexity gap between model-based RL ($\mathsf{AC}^0$) and model-free RL (NP-complete) persists in NP MDPs. Consequently, we have extended the results for 3-SAT MDP in Theorem 3.3 to a more general setting as desired. Similar explanations for Theorem 3.8 can be provided, akin to Remarks 3.4, 3.5, and 3.6 for 3-SAT MDPs, but we omit these to avoid repetition.

# 4 The Separation between Policy-based RL and Value-based RL

In Section 3, we demonstrate the representation complexity gap between model-based RL and model-free RL. In this section, our focus shifts to exploring the representation complexity hierarchy within model-free RL, encompassing policy-based RL and value-based RL. More specifically, we construct a broad class of MDPs where both the underlying model and the optimal policy are easy to represent, while the optimal value function is hard to represent. This further illustrates the representation hierarchy between different categories of RL algorithms.

## 4.1 CVP MDP

We begin by introducing the CVP MDPs, whose construction is rooted in the circuit value problem (CVP). The CVP involves computing the output of a given Boolean circuit (refer to Definition B.2) on a given input.

**Definition 4.1** (CVP MDP). An $n$-dimensional CVP MDP is defined as follows. Let $\mathcal{C}$ be the set of all circuits of size $n$. The state space $\mathcal{S}$ is defined by $\mathcal{S} = \mathcal{C} \times \{0, 1, \texttt{Unknown}\}^n$, where each state $s$ can be represented as $s = (\mathbf{c}, \mathbf{v})$. Here, $\mathbf{c}$ is a circuit consisting of $n$ nodes with $\mathbf{c}[i] = (\mathbf{c}[i][1], \mathbf{c}[i][2], g_i)$ describing the $i$-th node, where $\mathbf{c}[i][1]$ and $\mathbf{c}[i][2]$ indicate the input node and $g_i$ denotes the type of gate (including $\wedge, \vee, \neg, 0, 1$). When $g_i \in \{\wedge, \vee\}$, the outputs of $\mathbf{c}[i][1]$-th node and $\mathbf{c}[i][2]$-th node serve as the inputs; and when $g_i = \neg$, the output of $\mathbf{c}[i][1]$-th node serves as the input and $\mathbf{c}[i][2]$ is meaningless. Moreover, the node type of $0$ or $1$ denotes that the corresponding node is a leaf node with a value of $0$ or $1$, respectively, and therefore, $\mathbf{c}[i][1], \mathbf{c}[i][2]$ are both meaningless. The vector $\mathbf{v} \in \{0, 1, \texttt{Unknown}\}^n$ represents the value of the $n$ nodes, where the value $\texttt{Unknown}$ indicates that the value of this node has not been computed and is presently unknown. The action space is $\mathcal{A} = [n]$

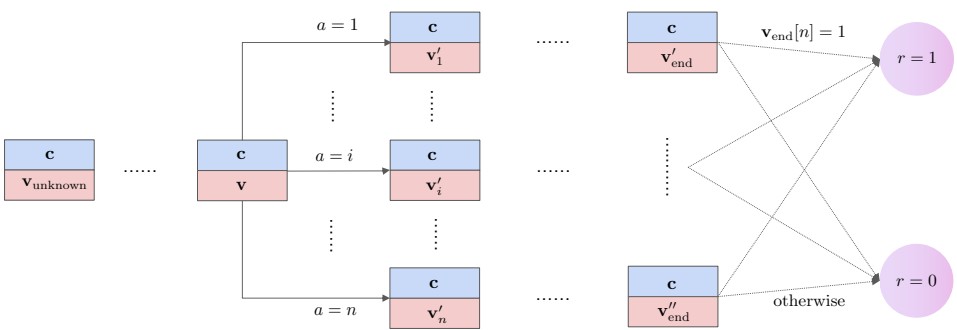

Figure 2: A visualization of CVP MDPs. Here, $\mathbf{v}_{\text{unknown}}$, which contains $n$ `Unknown` values, is the initial value vector. For any state $s$ including a circuit $\mathbf{c}$ and a value vector $\mathbf{v}$, choosing the action $i$, the environment transits to $(\mathbf{c}, \mathbf{v}'_i)$. Moreover, $\mathbf{v}_{\text{end}}$, $\mathbf{v}'_{\text{end}}$, and $\mathbf{v}''_{\text{end}}$ are value vectors at the end of the episode.

and the planning horizon is $H = n + 1$. Given a state-action pair $(s = (\mathbf{c}, \mathbf{v}), a)$, its reward $r(s, a)$ is given by:

$$r(s,a) = \begin{cases} 1 & \text{If the value of the output gate } \mathbf{v}[n] = 1, \\ 0 & \text{Otherwise.} \end{cases}$$

Moreover, the transition kernel is deterministic and can be defined as: $\mathcal{P}((\mathbf{c}, \mathbf{v}), a) = (\mathbf{c}, \mathbf{v}')$. Here, $\mathbf{v}'$ is obtained from $\mathbf{v}$ by computing and substituting the value of node $a$. More exactly, if the inputs of node $a$ have been computed, we can compute the output of the node $a$ and denote it as $\mathbf{o}[a]$. Then we have

$$\mathbf{v}'[j] = \begin{cases} \mathbf{v}[j] & \text{If } a \neq j, \\ \mathbf{o}[a] & \text{If } a = j \text{ and inputs of } a \text{ are computed,} \\ \texttt{Unknown} & \text{If } a = j \text{ and inputs of } a \text{ are not computed.} \end{cases}$$

Given a circuit $\mathbf{c}$, the initial state of CVP MDP is $(\mathbf{c}, \mathbf{v}_{\text{unknown}})$ where $\mathbf{v}_{\text{unknown}}$ denotes the vector containing $n$ `Unknown` values.

The visualization of CVP MDPs is given in Figure 2. In simple terms, each state $s = (\mathbf{c}, \mathbf{v})$ comprises information about a given size-$n$ circuit $\mathbf{c}$ and a vector $\mathbf{v} \in \{0, 1, \texttt{Unkown}\}^n$. At each step, the agent takes an action $a \in [n]$. If the $a$-th node has not been computed, and the input nodes are already computed, then the transition kernel of the CVP MDP modifies $\mathbf{v}[a]$ based on the type of gate $\mathbf{c}[a]$. The agent achieves the maximum reward of 1 only if it transforms the initial vector $\mathbf{v}_{\text{unknown}}$, consisting of $n$ `Unknown` values, into the $\mathbf{v}$ satisfying $\mathbf{v}[n] = 1$. This also indicates that CVP MDPs exhibit the capacity to model many real-world goal-conditioned problems and scenarios featuring sparse rewards. Hence, we have strategically encoded the circuit value problem into the CVP MDP in this manner. The representation complexity guarantee for the CVP MDP is provided below, and the proof is provided in Appendix F.1.

**Theorem 4.2** (Representation Complexity of CVP MDP). Let $\mathcal{M}_n$ be the $n$-dimensional CVP MDP defined in Definition 4.1. The reward function $r$, transition kernel $\mathcal{P}$, and optimal policy $\pi^*$ of $\mathcal{M}_n$ can be computed by circuits with polynomial size (in $n$) and constant depth, falling within the circuit complexity class $\mathsf{AC}^0$. However, the problem of computing the optimal value function $Q_1^*$ of $\mathcal{M}_n$ is P-complete under the log-space reduction.

Theorem 4.2 illustrates that, for CVP MDPs, the representation complexity of the optimal value function is notably higher than that of the underlying model and optimal policy.

**Remark 4.3.** We explain why P-completeness is considered challenging. In computational complexity theory, problems efficiently solvable in parallel fall into class NC. It is widely believed that P-complete problems cannot be efficiently solved in parallel (i.e., $\mathsf{NC} \neq \mathsf{P}$). Neural networks like MLP and Transformers [47], which are implemented in a highly parallel manner, face limitations when addressing P-complete problems. See Section 5 for details.

## 4.2 P MDP: A Broader Class of MDPs

We broaden the scope of CVP MDP to encompass a broader class of MDPs — P MDPs. In this extension, we can encode *any* P-complete problem into the MDP structure while preserving the results established for CVP MDP in Theorem 4.2. We introduce the P MDP as follows.

**Definition 4.4** (P MDP). Given a language $\mathcal{L}$ in P, and a circuit family $\mathcal{C}$, where $\mathcal{C}_n \in \mathcal{C}$ contains the circuits capable of recognizing strings of the length $n$ in $\mathcal{L}$. The size of the circuits in $\mathcal{C}_n$ is upper bounded by a polynomial $P(n)$. An $n$-dimensional P MDP based on $\mathcal{L}$ is defined as follows. The state space $\mathcal{S}$ is defined by $\mathcal{S} = \{0,1\}^n \times \mathcal{C}_n \times \{0,1,\texttt{Unknown}\}^{P(n)}$, where each state $s$ can be represented as $s = (\mathbf{x}, \mathbf{c}, \mathbf{v})$. Here, $\mathbf{c}$ is the circuit recognizing the strings of length $n$ in $\mathcal{L}$ with $\mathbf{c}[i] = (\mathbf{c}[i][1], \mathbf{c}[i][2], g_i)$ representing the $i$-th node where the output of nodes $i_1$ and $i_2$ serves as the input, and $g_i$ is the type of the gate (including $\wedge, \vee, \neg$, and $\texttt{Input}$). When $g_i \in \{\wedge, \vee\}$, the outputs of $\mathbf{c}[i][1]$-th node and $\mathbf{c}[i][2]$-th node serve as the inputs; and when $g_i = \neg$, the output of $\mathbf{c}[i][1]$-th node serves as the input and $\mathbf{c}[i][2]$ is meaningless. Moreover, the type $\texttt{Input}$ indicates that the corresponding node is the $\mathbf{c}[i][1]$-th bit of the input string. The vector $\mathbf{v} \in \{0,1,\texttt{Unknown}\}^{P(n)}$ representing the value of the $n$ nodes, and the value $\texttt{Unknown}$ indicates that the value of the corresponding node has not been computed, and hence is currently unknown. The action space is $\mathcal{A} = [P(n)]$ and the planning horizon is $H = P(n) + 1$. The reward of any state-action $(s = (\mathbf{x}, \mathbf{c}, \mathbf{v}), a)$ is defined by:

$$r(s,a) = \begin{cases} 1 & \text{If the value of the output gate } \mathbf{v}[P(n)] = 1, \\ 0 & \text{Otherwise.} \end{cases}$$

Moreover, the transition kernel is deterministic and can be defined as: $\mathcal{P}\big((\mathbf{x}, c, \mathbf{v}), a\big) = (\mathbf{x}, c, \mathbf{v}')$, where $\mathbf{v}'$ is obtained from $\mathbf{v}$ by computing and substituting the value of node $a$. In particular, if the inputs of node $a$ have been computed or can be read from the input string, we can determine the output of node $a$ and denote it as $\mathbf{o}[a]$. This yields the formal expression of $\mathbf{v}'$:

$$\mathbf{v}'[j] = \begin{cases} \mathbf{v}[j] & \text{If } a \neq j, \\ \mathbf{o}[a] & \text{If } a = j \text{ and inputs of } a \text{ are computed,} \\ \texttt{Unknown} & \text{If } a = j \text{ and inputs of } a \text{ are not computed.} \end{cases}$$

Given an input $\mathbf{x}$ and a circuit $\mathbf{c}$ capable of recognizing strings of specific length in $\mathcal{L}$, the initial state of P MDP is $(\mathbf{c}, \mathbf{v}_{\text{unknown}})$ where $\mathbf{v}_{\text{unknown}}$ denotes the vector containing $P(n)$ $\texttt{Unknown}$ values and $P(n)$ is the size of $\mathbf{c}$.

In the definition of P MDPs, we employ circuits to recognize the P-complete language $\mathcal{L}$ instead of using a Turing Machine, as done in the NP MDP in Definition 3.7. While it is possible to define P MDPs using a Turing Machine, we opt for circuits to maintain consistency with CVP MDP and facilitate our proof. Additionally, we remark that employing circuits to define NP MDPs poses challenges, as it remains elusive whether polynomial circuits can recognize NP-complete languages.

**Theorem 4.5** (Representation complexity of P MDP). For any P-complete language $\mathcal{L}$, consider its corresponding ($n$-dimensional) P MDP $\mathcal{M}_n$ as defined in Definition 4.4. The reward function $r$, transition kernel $\mathcal{P}$, and the optimal policy $\pi^*$ of $\mathcal{M}_n$ can be computed by circuits with polynomial size (in $n$) and constant depth, falling within the circuit complexity class $\mathsf{AC}^0$. However, the problem of computing the optimal value function $Q_1^*$ of $\mathcal{M}_n$ is P-complete under the log-space reduction.

The proof of Theorem 4.5 is deferred to Appendix F.2. Theorem 4.5 significantly broadens the applicability of Theorem 4.2 by enabling the encoding of any P-complete problem into the MDP structure, as opposed to a specific circuit value problem. In these expanded scenarios, the representation complexity of the model and optimal policy remains noticeably lower than that of the optimal value function.

Consequently, by combining Theorems 3.3, 3.8, 4.2, and 4.5, a potential representation complexity hierarchy has been unveiled. Specifically, the underlying model is the easiest to represent, followed by the optimal policy, with the optimal value exhibiting the highest representation complexity.

## 5  Connections to Deep RL

While we have uncovered the representation complexity hierarchy between model-based RL, policy-based RL, and value-based RL through the lens of computational complexity in Sections 3 and 4, these results offer limited insights for modern deep RL, where models, policies, and values are approximated by neural networks. To address this limitation, we further substantiate our revealed representation complexity hierarchy among different RL paradigms through the perspective of the expressiveness of neural networks. Specifically, we focus on the MLP with Rectified Linear Unit (ReLU) as the activation function[5] — an architecture predominantly employed in deep RL algorithms.

---

[5]Our results in this section are ready to be extended to other activation functions, such as Exponential Linear Unit (ELU), Gaussian Error Linear Unit (GeLU) and so on.

**Definition 5.1** (Log-precision MLP). An $L$-layer MLP is a function from input $\mathbf{e}_0 \in \mathbb{R}^d$ to output $\mathbf{y} \in \mathbb{R}^{d_y}$, recursively defined as

$$\mathbf{h}_1 = \text{ReLU}\left(\mathbf{W}_1 \mathbf{e}_0 + \mathbf{b}_1\right), \mathbf{W}_1 \in \mathbb{R}^{d_1 \times d}, \mathbf{b}_1 \in \mathbb{R}^{d_1},$$

$$\mathbf{h}_\ell = \text{ReLU}\left(\mathbf{W}_\ell \mathbf{h}_{\ell-1} + \mathbf{b}_\ell\right), \quad \mathbf{W}_\ell \in \mathbb{R}^{d_\ell \times d_{\ell-1}}, \mathbf{b}_\ell \in \mathbb{R}^{d_\ell},$$

$$\mathbf{y} = \mathbf{W}_L \mathbf{h}_L + \mathbf{b}_L, \mathbf{W}_L \in \mathbb{R}^{d_y \times d_L}, \mathbf{b}_L \in \mathbb{R}^{d_y},$$

where $2 \leq \ell \leq L - 1$ and $\text{ReLU}(x) = \max\{0, x\}$ for any $x \in \mathbb{R}$ is the standard ReLU activation. *Log-precision MLPs* refer to MLPs whose internal neurons can only store floating-point numbers within $O(\log n)$ bit precision, where $n$ is the maximal length of the input dimension.

The log-precision MLP is closely related to practical scenarios where the precision of the machine (e.g., 16 bits or 32 bits) is generally much smaller than the input dimension (e.g., 1024 or 2048 for the representation of image data). In our paper, all occurrences of MLPs will implicitly refer to the log-precision MLP, and we may omit explicit emphasis on log precision for the sake of simplicity. See Appendix B.3 for more details regarding log precision.

To employ MLPs to represent the model, policy, and value function, we encode each state $s$ and action $a$

|  | Transition | Reward | Optimal Policy | Optimal Value |
|---|---|---|---|---|
| Input | $(\mathbf{e}_s, \mathbf{e}_a)$ | $(\mathbf{e}_s, \mathbf{e}_a)$ | $\mathbf{e}_s$ | $(\mathbf{e}_s, \mathbf{e}_a)$ |
| Output | $\mathbf{e}_{s'}$ | $r(s,a)$ | $\mathbf{e}_a$ | $Q_1^*(s,a)$ |

Table 2: The input and output of the MLPs that represent the model, optimal policy, and optimal value function.

into embeddings $\mathbf{e}_s \in \mathbb{R}^{d_s}$ and $\mathbf{e}_a \in \mathbb{R}^{d_a}$, respectively. The detailed constructions of state embeddings and action embeddings of each type of MDPs are provided in Appendix G.1. In addition, we specify the input and output of MLPs that represent the model, policy, and value function in Table 2.

We now unveil the hierarchy of representation complexity from the perspective of MLP expressiveness. To begin with, we demonstrate the representation complexity gap between model-based RL and model-free RL.

**Theorem 5.2.** The reward function $r$ and transition kernel $\mathcal{P}$ of $n$-dimensional 3-SAT MDP and NP MDP can be represented by an MLP with constant layers, polynomial hidden dimension (in $n$), and ReLU as the activation function.

**Theorem 5.3.** Assuming that $\mathsf{TC}^0 \neq \mathsf{NP}$, the optimal policy $\pi_1^*$ and optimal value function $Q_1^*$ of $n$-dimensional 3-SAT MDP and NP MDP defined with respect to an NP-complete language $\mathcal{L}$ cannot be represented by an MLP with constant layers, polynomial hidden dimension (in $n$), and ReLU as the activation function.

The proof of Theorems 5.2 and 5.3 are deferred to Appendices G.2 and G.3, respectively. Theorems 5.2 and 5.3 show that the underlying model of 3-SAT MDP and NP MDP can be represented by constant-layer perceptron, while the optimal policy and optimal value function cannot. These demonstrate the representation complexity gap between model-based RL and model-free RL from the perspective of MLP expressiveness. The following two theorems further illustrate the representation complexity gap between policy-based RL and value-based RL, and the proof are deferred to Appendices G.4 and G.5, respectively.

**Theorem 5.4.** The reward function $r$, transition kernel $\mathcal{P}$, and optimal policy $\pi^*$ of $n$-dimensional CVP MDP and P MDP can be represented by an MLP with constant layers, polynomial hidden dimension (in $n$), and ReLU as the activation function.

**Theorem 5.5.** Assuming that $\mathsf{TC}^0 \neq \mathsf{P}$, the optimal value function $Q_1^*$ of $n$-dimensional CVP MDP and P MDP defined with respect to a P-complete language $\mathcal{L}$ cannot be represented by an MLP with constant layers, polynomial hidden dimension (in $n$), and ReLU as the activation function.

Combining Theorems 5.2, 5.3, 5.4, and 5.5, we reaffirm the potential representation complexity hierarchy uncovered in Sections 3 and 4 from the perspective of MLP expressiveness. To our best knowledge, this is the first result on representation complexity in RL from the perspective of MLP expressiveness, aligning more closely with modern deep RL and providing valuable insights for practice.

**Remark 5.6.** The results presented in this section underscore the importance of establishing NP-completeness and P-completeness in Sections 3 and 4. Specifically, constant-layer MLPs with polynomial hidden dimension are unable to simulate P-complete problems and NP-complete problems under the assumptions that $\mathsf{TC}^0 \neq \mathsf{P}$ and $\mathsf{TC}^0 \neq \mathsf{NP}$, which are widely believed to be impossible. In contrast, it is noteworthy that MLPs with constant layers and polynomial hidden dimension can

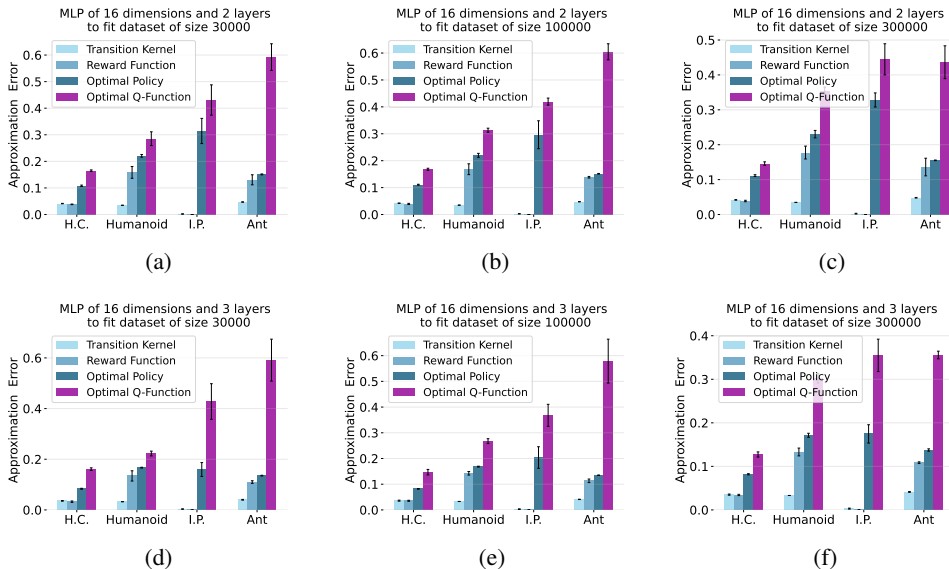

Figure 3: The approximation errors computed by employing MLPs with varying depths $d$ and widths $w$ to approximate the transition kernel, reward function, optimal policy, and optimal Q-function in four MuJoCo environments. In each subfigure, the title indicates the configuration including hidden dimensions, number of layers, and dataset size. The x-axis lists the four MuJoCo environments, where H.C. represents HalfCheetah and I.P. represents InvertedPendulum. The y-axis represents the approximation error defined in (D.1).

represent basic operations within $\mathsf{TC}^0$ (Lemma I.6), such as the Majority function. Consequently, the model, optimal policy, and optimal value function of "Majority MDPs" presented in [62] can be represented by constant-layer MLPs with polynomial size. Hence, the class of MDPs presented in [62] cannot demonstrate the representation complexity hierarchy from the lens of MLP expressiveness.

**Applicability and Extensions of Our Theory.** As mentioned in the introduction, our representation results have implications for the **statistical complexity** in RL, as detailed in Appendix C.1. Although we have shown that the revealed hierarchy of representation complexity holds for a wide range of MDPs in theory, examining its broader applicability is essential. We discuss **more general theoretical insights** and **extension to Transformer [47] architecture** to Appendices C.2 and C.3.

**Experiments.** We want to emphasize that our theoretical results do not apply to all MDPs, such as the MDP with all zero rewards and complex transitions. However, these additional MDP classes may not be typical in practice and could be considered pathological examples from a theoretical standpoint. To demonstrate that our theory captures practical problems, we conduct an empirical investigation into the representation complexity of different RL paradigms across various MuJoCo Gym environments [7]. Our empirical findings align with our theoretical conclusions. We report part of our experimental results in Figure 3, more detailed experimental description and results are deferred to Appendix D.

## 6 Conclusions

This paper studies three RL paradigms — model-based RL, policy-based RL, and value-based RL — from the perspective of representation complexity. Through leveraging computational complexity (including time complexity and circuit complexity) and the expressiveness of MLPs as representation complexity metrics, we unveil a potential hierarchy of representation complexity among different RL paradigms. Our theoretical framework posits that representing the model constitutes the most straightforward task, succeeded by the optimal policy, while representing the optimal value function poses the most intricate challenge. Our work contributes to a deeper understanding of the nuanced complexities inherent in various RL paradigms, providing valuable insights for the advancement of RL methodologies.

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

# A  Additional Related Works

**Representation Complexity in RL.**   In the pursuit of achieving efficient learning in RL, most existing works [e.g., 21, 42, 15, 23, 14, 50, 46, 17, 58, 25] adopt the (approximate) realizability assumption. This assumption allows the learner to have access to a function class that (approximately) captures the underlying model, optimal policy, or optimal value function, contingent upon the specific algorithm type in use. However, justifying the complexity of such a function class, with the capacity to represent the underlying model, optimal policy, or optimal value function, has remained largely unaddressed in these works. To the best of our knowledge, two exceptions are the works of [13] and [62], which consider the representation complexity in RL. As mentioned earlier, by using the number of linear pieces of piecewise linear functions and circuit complexity as metrics, these two works reveal that the representation complexity of the optimal value function surpasses that of the underlying model. Compared with these two works, our work also demonstrates the separation between model-based RL and value-based from multiple angles, including circuit complexity, time complexity, and the expressive power of MLP, where the last perspective seems completely new in the RL theory literature. Moreover, our result demonstrates a more significant separation between model-based RL and value-based RL. In addition, we also study the representation complexity of policy-based RL, providing a potential hierarchy among model-based RL, policy-based RL, and value-based RL from the above perspectives.

**Classic Computational Complexity Results.**   There are many classical computational complexity results of RL [37, 11, 32, 6]. These studies characterize the computational complexity of the process of solving the decision problems (finding the optimal decision) in RL. However, our work differs by examining the representation complexity hierarchy among different RL paradigms, using the computational complexity and expressiveness of MLPs as the complexity measure. Consequently, our findings do not contradict previous classical results and cannot be directly compared to them.

**Model-based RL, Policy-based RL, and Value-based RL.**   In the domain of RL, there are distinct paradigms that guide the learning process: model-based RL, policy-based RL, and value-based RL, each with its unique approach. In **model-based RL**, the primary objective of the learner is to estimate the underlying model of the environment and subsequently enhance the policy based on this estimated model. Most work in tabular RL [e.g., 19, 5, 55, 57, 56] fall within this category — they estimate the reward model and transition kernel using the empirical means and update the policy by performing the value iteration on the estimated model. Additionally, some works extend this approach to RL with linear function approximation [4, 61] and general function approximation [42, 17, 58, 51]. **Policy-based RL**, in contrast, uses direct policy updates to improve the agent's performance. Typical algorithms such as policy gradient [44], natural policy gradient [26], proximal policy optimization [38] fall into this category. A long line of works proposes policy-based algorithms with provable convergence guarantees and sample efficiency. See e.g., [33, 1, 2, 8, 39, 59, 9, 49, 48, 29, 60, 34, 40] and references therein. In **value-based RL**, the focus shifts to the approximation of the value function, and policy updates are driven by the estimated value function. A plethora of provable value-based algorithms exists, spanning tabular RL [22], linear RL [53, 24], and beyond [21, 14, 23, 58, 10, 35]. These works mainly explore efficient RL through the lens of sample complexity, with less consideration for representation complexity, which is the focus of our work.

# B  Additional Background Knowledge

## B.1  Function Approximation in Model-based, Policy-based, and Value-based RL

In modern reinforcement learning, we need to employ function approximation to solve complex decision-making problems. Roughly speaking, RL algorithms can be categorized into three types – model-based RL, policy-based RL, and value-based RL, depending on whether the algorithm aims to approximate the model, policy, or value. In general, policy-based RL and value-based RL can both be regarded as model-free RL, which represents a class of RL methods that do not require the estimation of a model. We assume the learner is given a function class $\mathcal{F}$, and we will specify the form of $\mathcal{F}$ in model-based RL, policy-based RL, and value-based RL, respectively.

- **Model-based RL:** the learner aims to approximate the model, including the reward function and the transition kernels. Specifically, $\mathcal{F} = \{(r : \mathcal{S} \times \mathcal{A} \mapsto [0,1], \mathcal{P} : \mathcal{S} \times \mathcal{A} \mapsto \Delta(\mathcal{S}))\}$. We also

want to remark that we consider the time-homogeneous setting, where the reward function and transition kernel are independent of the timestep $h$. For the time-inhomogeneous setting, we can choose $\mathcal{F} = \mathcal{F}_1 \times \cdots \times \mathcal{F}_H$ and let $\mathcal{F}_h$ approximate the reward and transition at the $h$-th step.

- **Policy-based RL:** the learner directly approximates the optimal policy $\pi^*$. The function class $\mathcal{F}$ takes the form $\mathcal{F} = \mathcal{F}_1 \times \cdots \times \mathcal{F}_h$ with $\mathcal{F}_h \subset \{\pi_h : \mathcal{S} \mapsto \Delta(\mathcal{A})\}$ for any $h \in [H]$.

- **Value-based RL:** the learner utilizes the function class $\mathcal{F} = \mathcal{F}_1 \times \cdots \times \mathcal{F}_H$ to capture the optimal value function $Q^*$, where $\mathcal{F}_h \subset \{Q_h : \mathcal{S} \times \mathcal{A} \mapsto [0, H]\}$ for any $h \in [H]$.

In previous literature [e.g., 23, 14, 50, 46, 17, 58, 25], a standard assumption is the *realizability* assumption – the ground truth model/optimal policy/optimal value is (approximately) realized in the given function class. Typically, a higher complexity of the function class leads to a larger sample complexity. Instead of focusing on sample complexity, as previous works have done, we are investigating how complex the function class should be by characterizing the *representation complexity* of the ground truth model, optimal policy, and optimal value function.

### B.2 Computational Complexity

To rigorously describe the representation complexity, we briefly introduce some background knowledge of classical computational complexity theory, and readers are referred to [3] for a more comprehensive introduction. We first define three classes of computational complexity classes P, NP, and L.

- P is the class of languages[6] that can be recognized by a *deterministic* Turing Machine in polynomial time.

- NP is the class of languages that can be recognized by a *nondeterministic* Turing Machine in polynomial time.

- L is the class containing languages that can be recognized by a deterministic Turing machine using a logarithmic amount of space.

To facilitate the readers, we also provide the definitions of deterministic Turing Machine and nondeterministic Turing Machine in Definition B.1.

**Definition B.1** (Turing Machine). A *deterministic Turing Machine (TM)* $M$ is described by a tuple $(\Gamma, \mathbb{Q}, \mathbb{T})$, where $\Gamma$ is the tape *alphabet* containing the "blank" symbol, "start" symbol, and the numbers 0 and 1; $\mathbb{Q}$ is a finite, non-empty set of states, including a start state $q_{\text{start}}$ and a halting state $q_{\text{halting}}$; and $\mathbb{T} : \mathbb{Q} \times \Gamma^k \mapsto \mathbb{Q} \times \Gamma^{k-1} \times \{\text{Left}, \text{Stay}, \text{Right}\}$, where $k \geq 2$, is the *transition function*, describing the rules $M$ use in each step. The only difference between a *nondeterministic Turing Machine (NDTM)* and a deterministic Turing Machine is that an NDTM has two transition functions $\mathbb{T}_0$ and $\mathbb{T}_1$, and a special state $q_{\text{accept}}$. At each step, the NDTM can choose one of two transitions to apply, and accept the input if there *exists* some sequence of these choices making the NDTM reach $q_{\text{accept}}$. A *configuration* of (deterministic or nondeterministic) Turing Machine $M$ consists of the contents of all nonblank entries on the tapes of $M$, the machine's current state, and the pointer on the tapes.

To quantify the representation complexity, we also adopt the circuit complexity, a fundamental concept in theoretical computer science, to characterize it. Circuit complexity focuses on representing functions as circuits and measuring the resources, such as the number of gates, required to compute these functions. We start with defining Boolean circuits.

**Definition B.2** (Boolean Circuits). For any $m, n \in \mathbb{N}_+$, a Boolean circuit $\mathbf{c}$ with $n$ inputs and $m$ outputs is a directed acyclic graph (DAG) containing $n$ nodes with no incoming edges and $m$ edges with no outgoing edges. All non-input nodes are called gates and are labeled with one of $\wedge$ (logical operation AND), $\vee$ (logical operation OR), or $\neg$ (logical operation NOT). The value of each gate depends on its direct predecessors. For each node, its fan-in number is the number of incoming edges, and its fan-out number is its outcoming edges. The size of $\mathbf{c}$ is the number of nodes in it, and the depth of $\mathbf{c}$ is the maximal length of a path from an input node to the output node. Without loss of generality, we assume the output node of a circuit is the final node of the circuit.

---

[6]Following the convention of computational complexity [3], we may use the term "language" and "decision problem" interchangeably.

A specific Boolean circuit can be used to simulate a function (or a computational problem) with a fixed number of input bits. When the input length varies, a sequence of Boolean circuits must be constructed, each tailored to handle a specific input size. In this context, circuit complexity investigates how the circuit size and depth scale with the input size of a given function. We provide the definitions of circuit complexity classes $\mathsf{AC}^0$ and $\mathsf{TC}^0$.

- $\mathsf{AC}^0$ is the class of circuits with constant depth, unbounded fan-in number, polynomial AND and OR gates.

- $\mathsf{TC}^0$ extends $\mathsf{AC}^0$ by introducing an additional unbounded-fan-in majority gate $\mathrm{MAJ}$, which evaluates to false when half or more arguments are false and true otherwise.

The relationship between the aforementioned five complexity classes is

$$\mathsf{AC}^0 \subsetneq \mathsf{TC}^0 \subset \mathsf{L} \subset \mathsf{P} \subset \mathsf{NP}.$$

The question of whether the relationship $\mathsf{TC}^0 \subset \mathsf{P} \subset \mathsf{NP}$ holds as a strict inclusion remains elusive in theoretical computer science. However, it is widely conjectured that $\mathsf{P} = \mathsf{NP}$ and $\mathsf{TC}^0 = \mathsf{P}$ are unlikely to be true.

**Uniformity of Circuits.** Given a circuit family $\mathcal{C}$, where $\mathbf{c}_i \in \mathcal{C}$ is the circuit takes $n$ bits as input, the uniformity condition is often imposed on the circuit family, requiring the existence of some possibly resource-bounded Turing machine that, on input $n$, produces a description of the individual circuit $\mathbf{c}_n$. When this Turing machine has a running time polynomial in $n$, the circuit family $\mathcal{C}$ is said to be P-uniform. And when this Turing machine has a space logarithmic in $n$, the circuit family $\mathcal{C}$ is said to be L-uniform.

### B.3 Log Precision.

In this work, we focus on MLPs, of which neuron values are restricted to be floating-point numbers of logarithmic (in the input dimension $n$) precision, and all computations operated on floating-point numbers will be finally truncated, similar to how a computer processes real numbers. Specifically, the log-precision assumption means that we can use $O(\log(n))$ bits to represent a real number, where the dimension of the input sequence is bounded by $n$. An important property is that it can represent all real numbers of magnitude $O(\mathrm{poly}(n))$ within $O(\mathrm{poly}(1/n))$ truncation error.

## C   More Discussions and Additional Restuls

### C.1   Connections to Statistical Complexity

To elaborate further on the connections to statistical/sample complexity, the previous sample complexity (in both online and offline RL) of finding an $\varepsilon$ optimal policy is typically $\mathrm{poly}(d, H, 1/\varepsilon) \cdot \log|\mathcal{H}|$, where $d$ represents the complexity measure in online RL (e.g., DEC in [17] and GEC in [58]) or the coverage coefficient in offline RL (e.g., [50] and [46]), $H$ denotes the horizon, and $\mathcal{H}$ stands for the model/policy/value hypothesis. According to our representation complexity hierarchy theory, the model-based hypothesis could be simpler since the ground truth model is easy to represent, resulting in a smaller $\log|\mathcal{H}|$. This provides an explanation of why model-based RL typically enjoys better sample efficiency than model-free RL. Furthermore, this connection highlights the importance of considering representation complexity in the design of sample-efficient RL algorithms.

We also remark that the planning error of computing the optimal policy and value function using the learned model is an **optimization error**, and is a parallel direction of statistical error (sample efficiency). In summary, we consider the **approximation error (representation complexity of the ground truth model/policy/value)** in our work and provide an implication for the **statistical error (sample efficiency of learning algorithms)**. We believe that exploring the twisted approximation error, optimization error, and statistical error, and providing a deeper comparison between model-based RL and model-free RL would be an interesting direction, and we will endeavor to explore this in our future work.

## C.2 Generality of Representation Complexity Hierarchy

First, we wish to underscore that our identified representation complexity hierarchy holds in a general way. Theoretically, our proposed MDPs can encompass a wide range of problems, as any NP or P problems can be encoded within their structure. More crucially, our thorough experiments in diverse simulated settings support the representation complexity hierarchy we have uncovered. In fact, we have a generalized result establishing a hierarchy between policy-based RL and value-based RL, as stated in the following proposition:

**Proposition C.1.** Given a Markov Decision Process (MDP) $\mathcal{M} = (\mathcal{S}, \mathcal{A}, H, \mathcal{P}, r)$, where $\mathcal{S} \subset \{0,1\}^n$ and $|\mathcal{A}| = O(\mathsf{poly}(n))$, the circuit complexity of the optimal value function will not fall below the optimal policy under the $\mathsf{TC}^0$ reduction.

*Proof.* Note that given the optimal value function $Q^*$, the optimal policy $\pi^*$ can be represented as $\pi^*(s) = \mathrm{argmax}_{a \in \mathcal{A}} Q(s,a)$, for any $s \in \mathcal{S}$. Therefore, we represent the optimal policy as the following Boolean circuits:

$$\pi^*(s) = \bigvee_{a \in \mathcal{A}} \left( a \wedge \left( \bigwedge_{a' \in \mathcal{A}} \mathbb{1}[Q^*(s,a) \geq Q^*(s,a')] \right) \right).$$

Therefore, the circuit complexity of the optimal value function will not fall below the optimal policy under the $\mathsf{TC}^0$ reduction. $\square$

However, our representation complexity hierarchy is not valid for all MDPs. For instance, in MDPs characterized by complex transition kernels and zero reward functions, the model's complexity surpasses that of the optimal policy and value function. However, these additional MDP classes may not be typical in practice and could be considered pathological examples from a theoretical standpoint. We leave the fully theoretical characterizing of representation hierarchy between model-based RL, policy-based RL, and value-based RL as an open problem. For instance, it could be valuable to develop a methodology for classifying MDPs into groups and assigning a complexity ranking to each group within our representation framework.

## C.3 Extension to Transformer Architectures

Our theoretical results can also naturally extend to the Transformer architectures. First, we formulate the Transformer architectures to represent the model, policy, and value function. We encode each state $s$ and action $a$ into a sequence $\mathbf{s}_s$ and $\mathbf{s}_a$, the detailed construction of the MDPs in this paper are listed as follows:

- **Sequences for the 3-SAT MDP.** The state of the $n$-dimensional 3-SAT MDP is denoted as $s = (\psi, \mathbf{v}, k)$. Here, $\psi$ is represented by a sequence of literals of length $3n$, and $\mathbf{v}$ by a sequence of length $n$. By concatenating these sequences, we represent the state $s = (\psi, \mathbf{v}, k)$ using a sequence $\boldsymbol{S}_s$ of length $4n + 1$. The action $a \in \{0,1\}$ is represented by a single token.

- **Sequences for the NP MDP.** The state of the $n$-dimensional NP MDP is $s = (c, k)$, where $c$ represents the configuration of a non-deterministic Turing machine (NTM). The configuration $c$ includes the machine's internal state $s_M$, the tape content $t$, and the tape head position $l$. These components are encoded as an integer, a sequence of length $P(n)$, and another integer, respectively. Thus, the state $s = (c, k)$ is represented by a sequence $\boldsymbol{S}_s$ of length $P(n)+3$. The action $a \in \{0,1\}$ is encoded with a single token.

- **Sequences for the CVP MDP.** The state of the $n$-dimensional CVP MDP is $s = (\mathbf{c}, \mathbf{v})$, where $\mathbf{c}$ denotes a circuit and $\mathbf{v}$ is a vector. Each node $\mathbf{c}[i]$ of the circuit is represented by a sequence of 3 tokens. Consequently, a sequence of length $3n$ represents the entire circuit $\mathbf{c}$, and the state $s = (\mathbf{c}, \mathbf{v})$ is encoded by a sequence $\boldsymbol{S}_s$ of length $4n$. The action $a \in [n]$ is represented by a single token.

- **Sequences for the P MDP.** The state of the $n$-dimensional P MDP is $s = (\mathbf{x}, \mathbf{c}, \mathbf{v})$. Assuming the circuit $\mathbf{c}$ has a size bounded by $P(n)$, we represent the circuit using a sequence of length $3P(n)$. The value vector $\mathbf{v}$ is encoded by a sequence of length $P(n)$, and the input string $\mathbf{x}$ by a sequence of length $n$. Therefore, the state $s = (\mathbf{x}, \mathbf{c}, \mathbf{v})$ is represented by the concatenated sequence $\boldsymbol{S}_s$. The action $a \in [P(n)]$ is encoded as a scalar.

With these formulations, we can extend our theoretical results to the Transformer architectures.

**Theorem C.2.** Assuming that $\mathsf{TC}^0 \neq \mathsf{NP}$, the optimal policy $\pi_1^*$ and optimal value function $Q_1^*$ of $n$-dimensional 3-SAT MDP and NP MDP defined with respect to an NP-complete language $\mathcal{L}$ cannot be represented by a Transformer with constant layers, polynomial hidden dimension (in $n$),and ReLU as the activation function.

**Theorem C.3.** Assuming that $\mathsf{TC}^0 \neq \mathsf{P}$, the optimal value function $Q_1^*$ of $n$-dimensional CVP MDP and P MDP defined with respect to a P-complete language $\mathcal{L}$ cannot be represented by a Transformer with constant layers, polynomial hidden dimension (in $n$), and ReLU as the activation function.

*Proof of Theorems C.2 and C.3.* According to the previous work [36], a Transformer with logarithmic precision, a fixed number of layers, and a polynomial hidden dimension can be simulated by a L-uniform $\mathsf{TC}^0$ circuit. On the other hand, the computation of the optimal policy and optimal value function for the 3-SAT MDP and NP MDP is NP-complete, and the computation of the optimal value function for CVP MDP and P MDP is P-complete. Therefore, the theorem holds under the assumption of $\mathsf{TC}^0 \neq \mathsf{NP}$ and $\mathsf{TC}^0 \neq \mathsf{P}$. □

**Theorem C.4.** The reward function $r$ and transition kernel $\mathcal{P}$ of $n$-dimensional 3-SAT MDP and NP MDP can be represented by a Transformer with constant layers, polynomial hidden dimension (in $n$), and ReLU as the activation function.

**Theorem C.5.** The reward function $r$, transition kernel $\mathcal{P}$, and optimal policy $\pi^*$ of $n$-dimensional CVP MDP and P MDP can be represented by a Transformer with constant layers, polynomial hidden dimension (in $n$), and ReLU as the activation function.

*Proof of Theorems C.4 and C.5.* According to previous works [16, 52], the Transformer model can aggregate the embeddings of the whole sequence to the embedding of one token with the attention mechanism. According to Theorems 5.2 and 5.4, an MLP with constant layers, polynomial hidden dimension (in $n$) and ReLU activation can represent these functions. Given an input sequence of states, the transformer first uses the attention layer to aggregate the whole sequence into a vector, and then just use the MLP module to calculate the corresponding functions. □

# D   Experimental Details

In this section, we empirically investigate the representation complexity of model-based RL, policy-based RL, and value-based RL and validate our theory with a comprehensive set of experiments on various common simulated environments. Following [62], fixing the depth $d$ and the width $w$, and denoting the class of MLPs with $d$ layers and $w$ hidden units (with input and output sizes adjusted to the context) as $\mathcal{F}$, we define the relative approximation errors as follows:

$$
\begin{aligned}
e_{\text{transition}} &= \min_{f \in \mathcal{F}} \frac{\mathbb{E}[\|f(s,a) - \mathcal{P}(s,a)\|^2]^{\frac{1}{2}}}{\mathbb{E}[\|\mathcal{P}(s,a) - \mathbb{E}[\mathcal{P}(s,a)]\|^2]^{\frac{1}{2}}}, \\
e_{\text{reward}} &= \min_{f \in \mathcal{F}} \frac{\mathbb{E}[(f(s,a) - r(s,a))^2]^{\frac{1}{2}}}{\mathbb{E}[(r(s,a) - \mathbb{E}[r(s,a)])^2]^{\frac{1}{2}}}, \\
e_{\text{policy}} &= \min_{f \in \mathcal{F}} \frac{\mathbb{E}[\|f(s) - \pi^*(s)\|^2]^{\frac{1}{2}}}{\mathbb{E}[\|\pi^*(s) - \mathbb{E}[\pi^*(s)]\|^2]^{\frac{1}{2}}}, \\
e_{\text{value}} &= \min_{f \in \mathcal{F}} \frac{\mathbb{E}[(f(s,a) - Q^*(s,a))^2]^{\frac{1}{2}}}{\mathbb{E}[(Q^*(s,a) - \mathbb{E}[Q^*(s,a)])^2]^{\frac{1}{2}}},
\end{aligned}
\tag{D.1}
$$

where the expectation is taken with respect to the distribution induced by the optimal policy, and the square root of mean squared errors is divided by the standard deviation to ensure that the scales of different errors match. Hence, the quantities $e_{\text{transition}}$, $e_{\text{reward}}$, $e_{\text{policy}}$, and $e_{\text{value}}$ defined in (D.1) characterize the difficulty for the MLP to approximate the transition kernel, the reward function, the optimal policy, and the optimal value function, respectively.

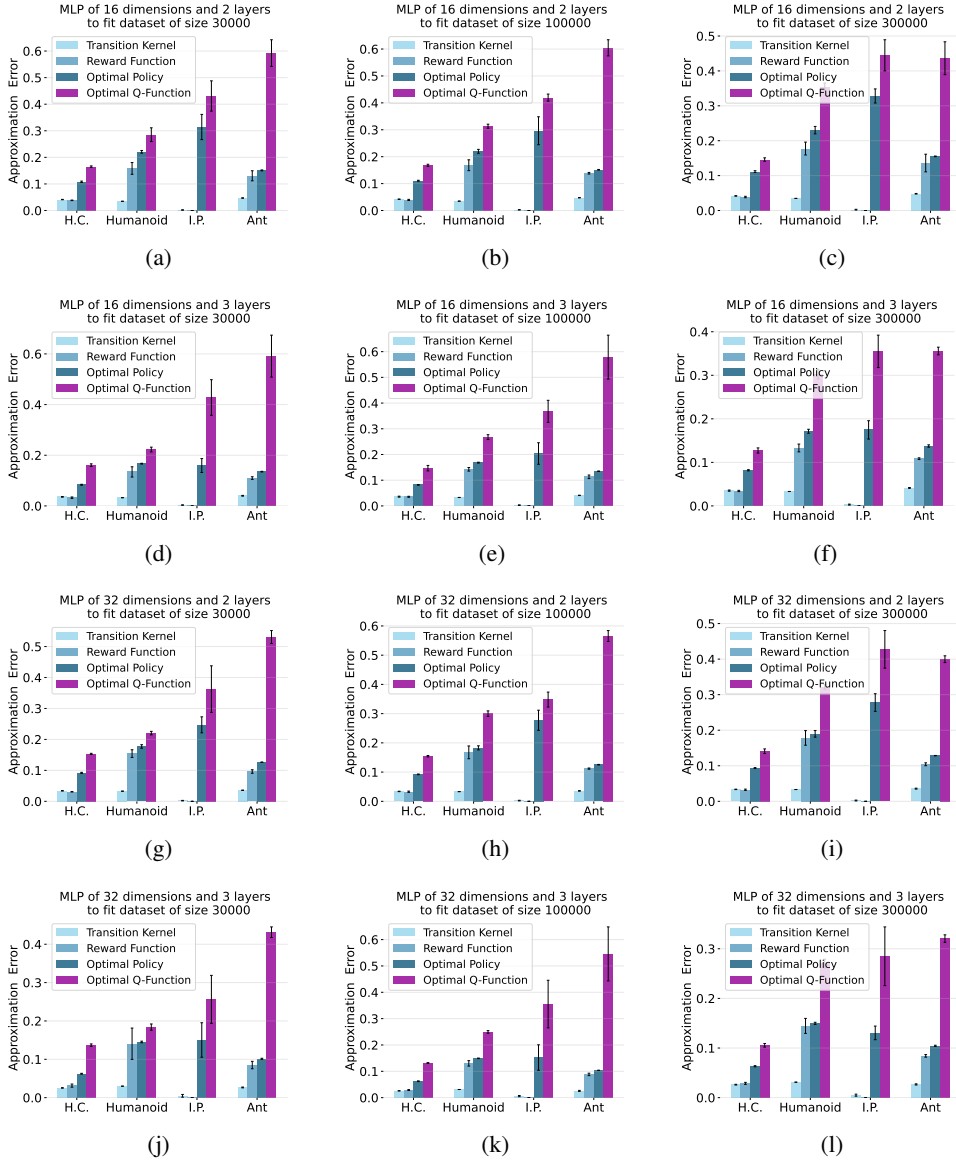

Figure 4: The approximation errors computed by employing MLPs with varying depths $d$ and widths $w$ to approximate the transition kernel, reward function, optimal policy, and optimal Q-function in four MuJoCo environments. In each subfigure, the title indicates the configuration including hidden dimensions, number of layers, and dataset size. The x-axis lists the four MuJoCo environments, where H.C. represents HalfCheetah and I.P. represents InvertedPendulum. The y-axis represents the approximation error defined in (D.1).

We conduct experiments on four MuJoCo Gym environments [7]: HalfCheetah-v4, Humanoid-v4, InvertedPendulum-v4, and Ant-v4. To calculate the approximation errors defined in (D.1), we employ the TD3 algorithm [18] to train a RL agent and utilize this agent to generate a dataset of size $\{30000, 100000, 300000\}$. The parameters used in TD3 are provided in Table 3. Subsequently, we use MLPs of different sizes (see Table 4) to approximate corresponding functions and calculate the approximation error. We run the experiment for each MLP configuration 5 times with 5 random seeds and report the means and standard deviations. All the experiments are conducted on an NVIDIA GeForce RTX 3090 Ti GPU. Figure 4 illustrates the results. In all experiments, the approximation error of the optimal value function is greater than the error of the optimal policy. And both of them are greater than the approximation error of the transition kernel and reward function. These results consistently indicate that the approximation errors of the optimal Q-function surpass those of the

optimal policy, which, in turn, exceed those of the transition kernel and reward functions, across all environments and configurations. These empirical results validate our representation hierarchy revealed from the theoretical perspective.

Table 3: Configurations for training the ground-truth agent by TD3 algorithm.

| | |
|---|---|
| Number of Layers of Actor-Network | 3 |
| Hidden Dimensions of Actor-Network | 256 |
| Number of Layers of Critic-Network | 3 |
| Hidden Dimensions of Critic-Network | 256 |
| Standard Deviation of Gaussian Exploration Noise | 0.1 |
| Discount Factor | 0.99 |
| Target Network Update Rate | 0.05 |

Table 4: Configurations for fitting the MLP to the corresponding functions and measuring the approximation error.

| | |
|---|---|
| Size of Dataset | $\{30000, 100000, 300000\}$ |
| Batch size | 128 |
| Optimization Steps | $\sim 70k$ |
| Number of Layers | $\{2, 3\}$ |
| Hidden Dimensions | $\{16, 32\}$ |
| Optimizer | Adam |
| Learning rate | 0.001 |

# E  Proofs for Section 3

## E.1  Proof of Theorem 3.3

*Proof of Theorem 3.3.* We investigate the representation complexity of the reward function, transition kernel, optimal value function, and optimal policy in sequence.

**Reward Function.** First, we prove the reward function can be implemented by $\mathsf{AC}^0$ circuits. Given a literal $\alpha \in \{u_1, u_2 \cdots, u_n, \neg u_1, \neg u_2, \cdots, \neg u_n\}$, we can obtain its value by

$$\alpha = \Big( \bigvee_{j \in [n]} (\mathbf{v}[j] \wedge \mathbb{1}[\alpha = u_i]) \Big) \vee \Big( \bigvee_{j \in [n]} (\neg \mathbf{v}[j] \wedge \mathbb{1}[\alpha = \neg u_i]) \Big).$$

After substituting the literal by its value under the assignment $\mathbf{v}$, we can calculate the 3-CNF Boolean formula $\psi(\mathbf{v})$ by two-layer circuits as its definition. Then the reward can be expressed as

$$r(s, a) = \mathbb{1}[\psi(\mathbf{v}) \wedge k = n + 1] + 0.5 \cdot \mathbb{1}[k = 2n + 2],$$

which further implies that the reward function can be implemented by $\mathsf{AC}^0$ circuits.

**Transition Kernel.** Then, we will implement the transition kernel by $\mathsf{AC}^0$ circuits. It is noted that we only need to modify the assignment $\mathbf{v}, k$. Given the input $\mathbf{v}, k$ and $a$, we have the output as follows:

$$\psi' = \psi, \qquad \mathbf{v}'[i] = (\mathbf{v}[i] \wedge \mathbb{1}[i \neq k]) \vee (a \wedge \mathbb{1}[i = k]),$$
$$k' = (k + 1) \cdot \mathbb{1}[k \geq 1] + \mathbb{1}[k = 0 \wedge a = 1] + (n + 2) \cdot \mathbb{1}[a = k = 0]. \tag{E.1}$$

It is noted that each element in the output is determined by at most $O(\log n)$ bits. Therefore, according to Lemma I.4, each bit of the output can be computed by two-layer circuits of polynomial size, and the overall output can be computed by $\mathsf{AC}^0$ circuits.

**Optimal Policy.** We aim to show that, given a state $s = (\psi, \mathbf{v}, k)$ as input, the problem of judging whether $\pi_1^*(s) = 1$ is NP-complete. We give a two-step proof.

**Step 1.** We first verify that this problem is in NP. Given a satisfiable assignment $\mathbf{v}^*$ as the certificate, we only need to verify the following things to determine whether there exists a sequence of actions to achieve the final reward of 1:

- The assignment is satisfiable;

- When $k \in [n]$ or $k = 0$, $\mathbf{v}[i] = \mathbf{v}^*[i]$, $i \in [k]$ and $\mathbf{v}^*[k+1] = 1$.

Notably, when exist such certificates, action $a = 1$ yields the reward of 1 and is consequently optimal. Conversely, when such certificates are absent, action $a = 1$ leads to the reward of 0, and in this case, $a = 0$ is the optimal action. Moreover, when $k = n + 1$ or $k = n + 2$, selecting $a = 1$ is always optimal. Consequently, given the certificates, we can verify whether $\pi_1^*(\psi, \mathbf{0}_n, 0) = 1$.

**Step 2.** Meanwhile, according to the well-known Cook-Levin theorem (Lemma I.1), the 3-SAT problem is NP-complete. Thus, our objective is to provide a polynomial time reduction from the 3-SAT problem to the problem of computing the optimal policy of 3-SAT MDP. Given a Boolean formula of length $n$, the number of variables is at most $n$. Then, we can pad several meaningless clauses such as $(u_1 \vee \neg u_1 \vee u_1)$ to obtain the 3-CNF Boolean formula $\psi'$ with $n$ clauses. Then the Boolean formula $\psi$ is satisfiable if and only if $\pi_1^*(\psi', \mathbf{0}_n, 0) = 1$. This provides a desired polynomial time reduction.

**Optimal Value Function.** To show the NP-completeness of computing the optimal value function, we formulate the decision version of this problem: given a state $s = (\psi, \mathbf{v}, k)$, an action $a$ and a number $\gamma$ as input, and the goal is to determine whether $Q_1^*(s, a) > \gamma$. According to the definition of NP-completeness, we need to prove that this problem belongs to the complexity class NP and then provide a polynomial-time reduction from a recognized NP-complete problem to this problem. These constitute the objectives of the subsequent two steps.

**Step 1.** We first verify that this problem is in NP. Given the input state $s = (\psi, \mathbf{v}, k)$, input action $a$, and input real number $\gamma$, we use the assignments $\mathbf{v}'$ of $\psi$ as certificates. When $\gamma \geq 1$, the verifier Turing Machine will reject the inputs, and when $\gamma < 0$, the verifier Turing Machine will accept the inputs. When $\gamma \in [0.5, 1)$, the verifier Turing Machine will accept the input when there exists a $\mathbf{v}'$ satisfying the following two conditions:

- The assignment $\mathbf{v}'$ is satisfiable for $\psi$

- When $k \in [n]$, it holds that $\mathbf{v}[i] = \mathbf{v}'[i]$ for all $i \in [k]$.

When $\gamma \in [0, 0.5)$, except in the scenario where the aforementioned two conditions are met, the verifier Turing Machine will additionally accept the input when $k = a = 0$. Then we have $Q_1^*(s, a) > \gamma$ if and only if there exists a certificate $\mathbf{v}'$ such that the verifier Turing Machine accepts the input containing $s, a, \gamma$, and the assignment $\mathbf{v}'$. Moreover, the verifier Turing Machine runs in at most polynomial time. Therefore, this problem is in NP.

**Step 2.** Meanwhile, according to the well-known Cook-Levin theorem(Lemma I.1), the 3-SAT problem is NP-complete. Thus, our objective is to provide a polynomial time reduction from the 3-SAT problem to the computation of the optimal value function for the 3-SAT MDP. Given a Boolean formula of length $n$, the number of variables is at most $n$. Then, we can pad several meaningless clauses such as $(u_1 \vee \neg u_1 \vee u_1)$ to obtain the 3-CNF Boolean formula $\psi'$ with $n$ clauses. The Boolean formula $\psi$ is satisfiable if and only if $Q_1^*((\psi', \mathbf{0}_n, 0), 1) > \frac{3}{4}$, which gives us a desired polynomial time reduction.

Combining these two steps, we can conclude that computing the optimal value function is NP-complete. $\qquad\square$

## E.2 Proof of Theorem 3.8

*Proof of Theorem 3.8.* Following the proof paradigm of Theorem 3.3, we characterize the representation complexity of the reward function, transition kernel, optimal policy, and optimal value function in sequence.

**Reward Function.** Given the state $s = (s_M, \mathbf{t}, l, k)$, the output of the reward

$$r(s, a) = \mathbb{1}[s_M = s_{\text{accept}} \wedge k = P(n) + 1] + 0.5 \cdot \mathbb{1}[k = 2P(n) + 2]. \tag{E.2}$$

It is not difficult to see that the reward function can be implemented by $\mathsf{AC}^0$ circuits.

**Transition Kernel.** We establish the representation complexity of the transition kernel by providing the computation formula for each element of the transited state. Our proof hinges on the observation that, for a state $s = (s_M, \mathbf{t}, l, k)$, we can extract the content $x$ of the location $l$ on the tape by the following formula:

$$\chi = \bigwedge_{i \in [P(n)]} (\mathbb{1}[i = l] \vee \mathbf{t}[i])$$

It is noted that we assume the contents written on the tape are $0$ and $1$. However, for the general case, we can readily extend the formula by applying it to each bit of the binary representation of the contents. Regarding the configuration $c' = (s'_M, \mathbf{t}', l')$ defined in (3.4), we observe that

(i) the Turing Machine state $s'_M$ is determined by $s_M, a$ and $\chi$;

(ii) the content of the location $l$ on the tape, $\mathbf{t}'[l]$, is determined by $s_M, a$ and $\chi$, whereas the contents of the other locations on the tape remain unaltered, i.e., $\mathbf{t}'[i] = \mathbf{t}[i]$ for $i \neq k$;

(iii) the pointer $l'$ is determined by $l, s_M, a$ and $\chi$.

Moreover, the number of steps $k'$ is determined by $k$ and $a$. Therefore, each element in the output is determined by at most $O(\log n)$ bits. According to Lemma I.4, each bit of the output can be computed by two-layer circuits of polynomial size, and the output can be computed by the $\mathsf{AC}^0$ circuits.

**Optimal Policy.** Our objective is to demonstrate the NP-completeness of the problem of determining whether $\pi_1^*(s) = 1$, given a state $s = (\psi, \mathbf{v}, k)$ as input. We will begin by establishing that this problem falls within the class NP, and subsequently, we will provide a polynomial-time reduction from the NP-complete language $\mathcal{L}$ to this specific problem.

**Step 1.** Given a sequence of choice of the branch as the certificate, we only need to verify the following two conditions to determine whether the optimal action $a = 1$:

- The final state of the Turing Machine $M$ is the accepted state.

- When $k \in [P(n)]$, the configuration of the Turing Machine after $k$-steps execution under the choice provided by the certificate is the same as the configuration in the current state.

- When $k \in [P(n)]$, the choice of the $k$-th step is branch 1.

Note that, when exist such certificates, action $a = 1$ can always get the reward of $1$ and is therefore optimal, and otherwise, action $a = 1$ always gets the reward of $0$, and $a = 0$ is always optimal. Moreover, when $k = P(n) + 1$ or $k = P(n) + 2$, we can always select the action $a = 1$ as the optimal action. So given the certificates, we can verify whether $\pi_1^*(\psi, \mathbf{0}_n, 0) = 1$.

**Step 2.** Given an input string $s_{\text{input}}$ of length $n$. We can simply get the initial configuration $c_0$ of the Turing Machine $M$ on the input $s_{\text{input}}$. Then $s_{\text{input}} \in \mathcal{L}$ if and only if $\pi_1^*((c_0, 0)) = 1$, which gives us a desired polynomial time reduction.

Combining these two steps, we know that computing the optimal policy of NP MDP is NP-complete.

**Optimal Value Function.** To facilitate our analysis, we consider the decision version of the problem of computing the optimal value function as follows: given a state $s = (s_M, \mathbf{t}, l, k)$, an action $a$, and a number $\gamma$ as input, and the goal is to determine whether $Q_1^*(s, a) > \gamma$.

**Step 1.** We first verify that the problem falls within the class NP. Given the input state $s$, we use a sequence of choice of the branch as the certificate. When $\gamma \geq 1$, the verifier Turing Machine will reject the inputs, and when $\gamma < 0$, the verifier Turing Machine will accept the input. When $\gamma \in [0.5, 1)$, the Turing Machine accepts the input when there is a certificate that satisfies the following two conditions:

- The final state of the Turing Machine $M$ is $s_{\text{accept}}$.

- When $k \in [P(n)]$, the configuration of the Turing Machine after $k$-steps execution under the choice provided by the certificate is the same as the configuration in the current state.

When $\gamma \in [0, 0.5)$, in the scenario where the aforementioned two conditions are met, the verifier Turing Machine will additionally accept the input when $k = a = 0$. Note that, all these conditions can be verified in polynomial time. Therefore, given the appropriate certificates, we can verify whether $Q_1^*(s, a) > \gamma$ in polynomial time.

**Step 2.** Given that $\mathcal{L}$ is NP-complete, it suffices to provide a polynomial time reduction from the $\mathcal{L}$ to the problem of computing the optimal value function of NP MDP. Let $s_{\text{input}}$ be an input string of length $n$. To obtain the initial configuration $c_0$ of the Turing Machine $M$ on the input $s_{\text{input}}$, we simply copy the input string onto the tape, set the pointer to the initial location, and designate the state of the Turing Machine as the initial state. Therefore, $s_{\text{input}} \in \mathcal{L}$ if and only if $Q_1^*((c_0, 0), 1) > \frac{3}{4}$, which provides a desired polynomial time.

Combining these two steps, we can conclude that computing the optimal value function of NP MDP is NP-complete. $\square$

# F Proofs for Section 4

## F.1 Proof of Theorem 4.2

*Proof of Theorem 4.2.* We characterize the representation complexity of the reward function, transition kernel, optimal policy, and optimal value function in sequence.

**Reward Function.** First, we prove that the reward function of CVP MDP can be computed by $\mathsf{AC}^0$ circuits. According to the definition, the output is

$$r(s, a) = \mathbb{1}[\mathbf{v}[n] = 1].$$

Therefore, the problem of computing the reward function falls within the complexity class $\mathsf{AC}^0$.

**Transition Kernel.** Then, we prove that the transition kernel of CVP MDP can be computed by $\mathsf{AC}^0$ circuits. Given the state-action pair $(s = (\mathbf{c}, \mathbf{v}), a)$, we denote the next state $\mathcal{P}(s, a)$ as $s' = (\mathbf{c}', \mathbf{v}')$. For any index $i \in [n]$, we can simply fetch the node $\mathbf{c}[i]$ and its value by

$$\mathbf{c}[i] = \bigvee_{j \in [n]} (\mathbf{c}[j] \wedge \mathbb{1}[i = j]), \qquad \mathbf{v}[i] = \bigvee_{j \in [n]} (\mathbf{v}[j] \wedge \mathbb{1}[i = j]), \tag{F.1}$$

where the AND and OR operations are bit-wise operations. Given the node $\mathbf{c}[a]$ and its inputs $\mathbf{v}[\mathbf{c}[a][1]]$ and $\mathbf{v}[\mathbf{c}[a][2]]$, we calculate the value of the $a$-th node and denote it as $\bar{\mathbf{o}}[a]$. Here, $\bar{\mathbf{o}}[a]$ represents the correct output of the $a$-th node when its inputs are computed, and is undefined when the inputs of the $a$-th node have not been computed. Therefore, let $\bar{\mathbf{o}}[a]$ be Unknown when the inputs of the $a$-th node have not been computed. Specifically, we can compute $\bar{\mathbf{o}}[a]$ as follows:

$$\begin{aligned}
\bar{\mathbf{o}}[a] = &\Big( \mathbb{1}[g_a = \wedge] \wedge \mathbb{1}[\mathbf{v}[\mathbf{c}[i][1]], \mathbf{v}[\mathbf{c}[i][2]] \neq \texttt{Unknown}] \wedge (\mathbf{v}[\mathbf{c}[i][1]] \wedge \mathbf{v}[\mathbf{c}[i][2]]) \Big) \\
&\vee \Big( \mathbb{1}[g_a = \vee] \wedge \mathbb{1}[\mathbf{v}[\mathbf{c}[i][1]], \mathbf{v}[\mathbf{c}[i][2]] \neq \texttt{Unknown}] \wedge (\mathbf{v}[\mathbf{c}[i][1]] \vee \mathbf{v}[\mathbf{c}[i][2]]) \Big) \\
&\vee \Big( \mathbb{1}[g_a = \neg] \wedge \mathbb{1}[\mathbf{v}[\mathbf{c}[i][1]] \neq \texttt{Unknown}] \wedge (\neg \mathbf{v}[\mathbf{c}[i][1]]) \Big) \vee \Big( \mathbb{1}[g_a \in \{0, 1\}] \wedge g_a \Big) \quad \text{(F.2)} \\
&\vee \Big( \texttt{Unknown} \wedge \big( (\mathbb{1}[g_a \in \{\vee, \wedge\}] \wedge \mathbb{1}[\texttt{Unknown} \in \{\mathbf{v}[\mathbf{c}[i][1]], \mathbf{v}[\mathbf{c}[i][2]]\}]) \vee \\
&\quad (\mathbb{1}[g_a = \neg] \wedge \mathbb{1}[\mathbf{v}[\mathbf{c}[i][1]]] = \texttt{Unknown}) \big) \Big).
\end{aligned}$$

Furthermore, we can express $s' = (\mathbf{c}', \mathbf{v}')$, the output of the transition kernel, as

$$\mathbf{c}' = \mathbf{c}, \qquad \mathbf{v}'[i] = (\bar{\mathbf{o}}[a] \wedge \mathbb{1}[i = a]) \wedge (\mathbf{v}[i] \wedge \mathbb{1}[i \neq a]),$$

which implies that the transition kernel of CVP MDP can be computed by $\mathsf{AC}^0$ circuits.

**Optimal Policy.** Based on our construction of CVP MDP, it is not difficult to see that $\pi_1^* = \pi_2^* = \cdots = \pi_H^*$. For simplicity, we omit the subscript and use $\pi^*$ to represent the optimal policy. Intuitively, the optimal policy is that given a state $s = (\mathbf{c}, \mathbf{v})$, we find the nodes with the smallest index among the nodes whose inputs have been computed and output has not been computed. Formally, this optimal policy can be expressed as follows. Given a state $s = (\mathbf{c}, \mathbf{v})$, denoting $\mathbf{c}[i] = (\mathbf{c}[i][1], \mathbf{c}[i][2], g_i)$, let $\mathcal{G}(s)$ be a set defined by:

$$
\begin{aligned}
\mathcal{G}(s) = & \{i \in [n] \mid g_i \in \{\wedge, \vee\}, \mathbf{v}[\mathbf{c}[i][1]], \mathbf{v}[\mathbf{c}[i][2]] \in \{0, 1\}, \mathbf{v}[i] = \texttt{Unknown}\} \\
& \cup \{i \in [n] \mid g_i = \neg, \mathbf{v}[\mathbf{c}[i][1]] \in \{0, 1\}, \mathbf{v}[i] = \texttt{Unknown}\} \\
& \cup \{i \in [n] \mid g_i \in \{0, 1\}, \mathbf{v}[i] = \texttt{Unknown}\}.
\end{aligned}
\tag{F.3}
$$

The set $\mathcal{G}(s)$ defined in (F.3) denotes the indices for which inputs have been computed, and the output has not been computed. Consequently, the optimal policy $\pi^*(s)$ is expressed as $\pi^*(s) = \min \mathcal{G}(s)$. If the output of the circuit $\mathbf{c}$ is 1, the policy $\pi^*$ can always get the reward 1, establishing its optimality. And if the output of circuit $\mathbf{c}$ is 0, the optimal value is 0 and $\pi^*$ is also optimal. Therefore, we have verified that the $\pi^* = \min \mathcal{G}(s)$ defined by us is indeed the optimal policy. Subsequently, we aim to demonstrate that the computational complexity of $\pi^*$ resides within $\mathsf{AC}^0$. Let $\Upsilon[i]$ denote the indicator of whether $i \in \mathcal{G}(s)$, i.e., $\Upsilon[i] = 1$ signifies that $i \in \mathcal{G}(s)$, while $\Upsilon[i] = 0$ denotes that $i \notin \mathcal{G}(s)$. According to (F.3), we can compute $\Upsilon[i]$ as follows:

$$
\begin{aligned}
\Upsilon[i] = & \big(\mathbb{1}[g_i \in \{\wedge, \vee\}] \wedge \mathbb{1}[\mathbf{v}[\mathbf{c}[i][1]], \mathbf{v}[\mathbf{c}[i][2]] \in \{0, 1\}] \wedge \mathbb{1}[\mathbf{v}[i] = \texttt{Unknown}]\big) \\
& \vee \big(\mathbb{1}[g_i = \neg] \wedge \mathbb{1}[\mathbf{v}[\mathbf{c}[i][1]] \in \{0, 1\}] \wedge \mathbb{1}[\mathbf{v}[i] = \texttt{Unknown}]\big) \\
& \vee \big(\mathbb{1}[g_i \in \{0, 1\}] \wedge \mathbb{1}[\mathbf{v}[i] = \texttt{Unknown}]\big).
\end{aligned}
$$

Under this notation, we arrive at the subsequent expression for the optimal policy $\pi^*$:

$$
\pi^*(s) = \bigvee_{i \in [n]} (\Upsilon'[i] \wedge i), \qquad \text{where } \Upsilon'[i] = \neg\Big(\bigvee_{j < i} \Upsilon[j]\Big) \wedge \Upsilon[i].
$$

Therefore, the computation complexity of the optimal policy falls in $\mathsf{AC}^0$.

**Optimal Value Function.** We prove that the computation of the value function of CVP MDP is P-complete under the log-space reduction. Considering that the reward in a CVP MDP is constrained to be either 0 or 1, we focus on the decision version of the optimal value function computation. Given a state $s = (\mathbf{c}, \mathbf{v})$ and an action $a$ as input, the objective is to determine whether $Q_1^*(s, a) = 1$. In the subsequent two steps, we demonstrate that this problem is within the complexity class P and offer a log-space reduction from a known P-complete problem (CVP problem) to this decision problem.

**Step 1.** We first verify the problem is in P. According to the definition, a state $s = (\mathbf{c}, \mathbf{v})$ can get the reward 1 if and only if the output of $\mathbf{c}$ is 1. A natural algorithm to compute the value of the circuit $\mathbf{c}$ is computing the values of nodes with the topological order. The algorithm runs in polynomial time, indicating that the problem is in P.

**Step 2.** Then we prove that the problem is P-complete under the log-space reduction. According to Lemma I.2, the CVP problem is P-complete. Thus, our objective is to provide a log-space reduction from the CVP problem to the computation of the optimal value function for the CVP MDP. Given a circuit $\mathbf{c}$ of size $n$ and a vector $\mathbf{v}_{\text{unkown}}$ containing $n$ $\texttt{Unknown}$ values, consider $s = (\mathbf{c}, \mathbf{v}_{\text{unkown}})$. Let $i = \pi^*(s)$, where the optimal policy $\pi^*$ is defined in the proof of the "optimal policy function" part. The output of $\mathbf{c}$ is 1 if and only if $Q_1^*(s, i) = 1$. Furthermore, the reduction is accomplished by circuits in $\mathsf{AC}^0$ and, consequently, falls within $\mathsf{L}$.

Combining these two steps, we know that computing the optimal value function is P-complete under the log-space reduction. $\qquad\square$

### F.2 Proof of Theorem 4.5

*Proof of Theorem 4.5.* We characterize the representation complexity of the reward function, transition kernel, optimal policy, and optimal value function in P MDPs in sequence.

**Reward Function.** We prove that the reward function of P MDP can be computed by $\mathsf{AC}^0$ circuits. According to the definition, the output is

$$
r(s, a) = \mathbb{1}[\mathbf{v}[P(n)] = 1].
$$

So the complexity of the reward function falls within the complexity class $\mathsf{AC}^0$.

**Transition Kernel.** First, we prove that the transition kernel of P MDP can be computed by $\mathsf{AC}^0$ circuits. Given a state-action pair $(s = (\mathbf{x}, \mathbf{c}, \mathbf{v}), a)$, we denote the next state $\mathcal{P}(s, a)$ by $s' = (\mathbf{x}', \mathbf{c}', \mathbf{v}')$. Similar to (F.1) in the proof of Theorem 4.2, we can fetch the node $\mathbf{c}[i]$, its value $\mathbf{v}[i]$, and the $i$-th character of the input string $\mathbf{x}[i]$. We need to compute the output of the $a$-th node. Given the node $\mathbf{c}[a]$ and its inputs $\mathbf{v}[\mathbf{c}[a][1]]$ and $\mathbf{v}[\mathbf{c}[a][2]]$ or $\mathbf{x}[\mathbf{c}[a][1]]$, we can compute the $a$-th node's value $\bar{\mathbf{o}}[a]$ similar to (F.2), where $\bar{\mathbf{o}}[a]$ is the correct output of the $a$-th node if the inputs are computed, and is $\texttt{Unknown}$ when the inputs of the $a$-th node contain the $\texttt{Unknown}$ value. In detail, $\bar{\mathbf{o}}$ can be computed as Here, $\bar{\mathbf{o}}[a]$ can be computed as:

$$
\begin{aligned}
\bar{\mathbf{o}}[a] =& \Big( \mathbb{1}[g_a = \wedge] \wedge \mathbb{1}[\mathbf{v}[\mathbf{c}[i][1]], \mathbf{v}[\mathbf{c}[i][2]] \neq \texttt{Unknown}] \wedge (\mathbf{v}[\mathbf{c}[i][1]] \wedge \mathbf{v}[\mathbf{c}[i][2]]) \Big) \\
& \vee \Big( \mathbb{1}[g_a = \vee] \wedge \mathbb{1}[\mathbf{v}[\mathbf{c}[i][1]], \mathbf{v}[\mathbf{c}[i][2]] \neq \texttt{Unknown}] \wedge (\mathbf{v}[\mathbf{c}[i][1]] \vee \mathbf{v}[\mathbf{c}[i][2]]) \Big) \\
& \vee \Big( \mathbb{1}[g_a = \neg] \wedge \mathbb{1}[\mathbf{v}[\mathbf{c}[i][1]] \neq \texttt{Unknown}] \wedge (\neg \mathbf{v}[\mathbf{c}[i][1]]) \Big) \\
& \vee \Big( \mathbb{1}[g_a = \texttt{Input}] \wedge \mathbf{x}[\mathbf{c}[i][1]] \Big) \\
& \vee \Big( \texttt{Unknown} \wedge \Big( (\mathbb{1}[g_a \in \{\vee, \wedge\}] \wedge \mathbb{1}[\texttt{Unknown} \in \{\mathbf{v}[\mathbf{c}[i][1]], \mathbf{v}[\mathbf{c}[i][2]]\}]) \vee \\
& \qquad\qquad (\mathbb{1}[g_a = \neg] \wedge \mathbb{1}[\mathbf{v}[\mathbf{c}[i][1]]] = \texttt{Unknown}) \Big) \Big).
\end{aligned}
\tag{F.4}
$$

Then the next state $s' = (\mathbf{x}', \mathbf{c}', \mathbf{v}')$ can be expressed as

$$
\mathbf{x}' = \mathbf{x}, \qquad \mathbf{c}' = \mathbf{c}, \qquad \mathbf{v}'[i] = (\mathbf{o}[a] \wedge \mathbb{1}[i = a]) \wedge (\mathbf{v}[i] \wedge \mathbb{1}[i \neq a]),
$$

which yields that the transition kernel of P MDP can be computed by $\mathsf{AC}^0$ circuits.

**Optimal Policy.** Given a state $s = (\mathbf{x}, \mathbf{c}, \mathbf{v})$, the optimal policy finds the nodes with the smallest index among the nodes whose inputs have been computed and output has not been computed. To formally define the optimal policy, we need to introduce the notation $\widetilde{\mathcal{G}}(s)$ to represent the set of indices of which inputs have been computed and output has not been computed. Given a state $s = (\mathbf{x}, \mathbf{c}, \mathbf{v})$, denoting $\mathbf{c}[i] = (\mathbf{c}[i][1], \mathbf{c}[i][2], g_i)$, the set $\widetilde{\mathcal{G}}(s)$ is defined by:

$$
\begin{aligned}
\widetilde{\mathcal{G}}(s) = & \{i \in [P(n)] \mid g_i \in \{\wedge, \vee\}, \mathbf{v}[\mathbf{c}[i][1]], \mathbf{v}[\mathbf{c}[i][2]] \in \{0, 1\}, \mathbf{v}[i] = \texttt{Unknown}\} \\
& \cup \{i \in [P(n)] \mid g_i = \neg, \mathbf{v}[\mathbf{c}[i][1]] \in \{0, 1\}, \mathbf{v}[i] = \texttt{Unknown}\} \\
& \cup \{i \in [P(n)] \mid g_i = \texttt{Input}, \mathbf{v}[i] = \texttt{Unknown}\}.
\end{aligned}
\tag{F.5}
$$

Under this notation, we can verify that the optimal policy is given by $\pi^*(s) = \min \widetilde{\mathcal{G}}(s)$. Here we omit the subscript and use $\pi^*$ to represent the optimal policy since $\pi_1^* = \pi_2^* = \cdots = \pi_H^*$. Specifically, (i) if the output of the circuit $c$ is 1, the policy $\pi^*$ can always get the reward 1 and is hence optimal; (ii) if the output of circuit $c$ is 0, the optimal value is 0 and $\pi^*$ is also optimal. Therefore, our objective is to prove that the computation complexity of $\pi^*$ falls in $\mathsf{AC}^0$. Let $\widetilde{\Upsilon}[i]$ be the indicator of whether $i \in \widetilde{\mathcal{G}}(s)$, i.e. $\widetilde{\Upsilon}[i] = 1$ indicates that $i \in \widetilde{\mathcal{G}}(s)$ and $\widetilde{\Upsilon}[i] = 0$ indicates $i \notin \widetilde{\mathcal{G}}(s)$. By the definition of $\widetilde{\mathcal{G}}(s)$ in (F.5), we can compute $\widetilde{\Upsilon}[i]$ as follows:

$$
\begin{aligned}
\widetilde{\Upsilon}[i] = & \big( \mathbb{1}[g_i \in \{\wedge, \vee\}] \wedge \mathbb{1}[\mathbf{v}[\mathbf{c}[i][1]], \mathbf{v}[\mathbf{c}[i][2]] \in \{0, 1\}] \wedge \mathbb{1}[\mathbf{v}[i] = \texttt{Unknown}] \big) \\
& \vee \big( \mathbb{1}[g_i = \neg] \wedge \mathbb{1}[\mathbf{v}[\mathbf{c}[i][1]] \in \{0, 1\}] \wedge \mathbb{1}[\mathbf{v}[i] = \texttt{Unknown}] \big) \\
& \vee \big( \mathbb{1}[g_i = \texttt{Input}] \wedge \mathbb{1}[\mathbf{v}[i] = \texttt{Unknown}] \big).
\end{aligned}
$$

Then, we can express the optimal policy as

$$
\pi^*(s) = \bigvee_{i \in [P(n)]} (\widetilde{\Upsilon}'[i] \wedge i) \qquad \text{where } \widetilde{\Upsilon}'[i] = \neg \Big( \bigvee_{j < i} \widetilde{\Upsilon}[j] \Big) \wedge \widetilde{\Upsilon}[i].
$$

Therefore, the computational complexity of the optimal policy falls in $\mathsf{AC}^0$.

**Optimal Value Function.** We prove that the computation of the value function of P MDP is P-complete under the log-space reduction. Note that the reward of the P MDP can be only 0 or 1, we

consider the decision version of the problem of computing the optimal value function as follows: given a state $s = (\mathbf{x}, \mathbf{c}, \mathbf{v})$ and an action $a$ as input, the goal is determining whether $Q_1^*(s, a) = 1$. we need to prove that this problem belongs to the complexity class P and then provide a log-space reduction from a recognized P-complete problem to this problem.

**Step 1.** We first verify the problem is in P. According to the definition, a state $s = (\mathbf{x}, \mathbf{c}, \mathbf{v})$ can get the reward 1 if and only if the output of $\mathbf{c}$ is 1. A natural algorithm to compute the value of the circuit $\mathbf{c}$ is computing the values of nodes according to the topological order. This algorithm runs in polynomial time, showing that the target decision problem is in P.

**Step 2.** Then we prove that the problem is P-complete under the log-space reduction. Under the condition that $\mathcal{L}$ is P-complete, our objective is to provide a log-space reduction from $\mathcal{L}$ to the computation of the optimal value function for the P MDP. By Lemma I.3, a language in P has log-space-uniform circuits of polynomial size. Therefore, there exists a Turing Machine that can generate a description of a circuit $\mathbf{c}$ in log-space which can recognize all strings of length $n$ in $\mathcal{L}$. Therefore, given any input string $\mathbf{x}$ of length $n$, we can find a corresponding state $s = (\mathbf{x}, \mathbf{c}, \mathbf{v}_{\text{unknown}})$, where $\mathbf{v}_{\text{unknown}}$ denotes the vector containing $P(n)$ Unknown values. Let $i = \pi^*(s)$, where $\pi^*$ is the optimal policy defined in the "optimal policy" proof part. Then $\mathbf{x} \in \mathcal{L}$ if and only if $Q_1^*(s, i) = 1$. This provides a desired log-space reduction. We also want to remark that here the size of reduction circuits in L should be smaller than $P(n)$. This condition can be easily satisfied since we can always find a sufficiently large polynomial $P(n)$.

Combining the above two steps, we can conclude that computing the optimal value function is P-complete. □

# G Proofs for Section 5

## G.1 State Embeddings and Action Embeddings

**Embeddings for the 3-SAT MDP.** The state of the $n$ dimension 3-SAT MDP $s = (\psi, \mathbf{v}, k)$. We can use an integer ranging from 1 to $2n$ to represent a literal from $\mathcal{V} = \{u_1, \neg u_1, \cdots, u_n, \neg u_n\}$. For example, we can use $(i, i + n)$ to represent $(u_i, \neg u_i)$ for any $i \in [n]$. Therefore, we can use a $3n$ dimensional vector $\mathbf{e}_\psi$ to represent the $\psi \in \mathcal{V}^{3n}$ and a $4n + 1$ dimensional vector $\mathbf{e}_s = (\mathbf{e}_\psi, \mathbf{v}, k)$ to represent the state $s = (\psi, \mathbf{v}, k)$. And we use a scalar to represent the action $a \in \{0, 1\}$.

**Embeddings for the NP MDP.** The state of the $n$-dimension NP MDP is denoted as $s = (c, k)$. A configuration of a non-deterministic Turing Machine, represented by $c$, encompasses the state of the Turing Machine $s_M$, the contents of the tape $t$, and the pointer on the tape $l$. To represent the state of the Turing Machine, the tape of the Turing Machine, and the pointer, we use an integer, a vector of $P(n)$ dimensions, and an integer, respectively. Therefore, a $P(n) + 2$ dimensional vector $\mathbf{e}_c$ is employed to represent the configuration, and a $P(n) + 3$ dimensional vector $\mathbf{e}_s = (\mathbf{e}_c, k)$ is used to represent the state $s = (c, k)$. Additionally, a scalar is utilized to represent the action $a \in \{0, 1\}$.

**Embeddings for the CVP MDP.** The state of the $n$-dimension CVP MDP is denoted as $s = (\mathbf{c}, \mathbf{v})$. Utilizing a 3 dimensional vector, we represent a node $\mathbf{c}[i]$ of the circuit. Consequently, a $3n$ dimensional vector $\mathbf{e}_\mathbf{c}$ is employed to represent the circuit $\mathbf{c}$, and a $4n$ dimensional vector $\mathbf{e}_s = (\mathbf{e}_\mathbf{c}, \mathbf{v})$ is used to represent the state $s = (\mathbf{c}, \mathbf{v})$. In this representation, an integer ranging from 1 to $n$ is used to signify the node index, an integer ranging from 1 to 5 is employed to denote the type of a node, and an integer ranging from 1 to 3 is utilized to represent the value of a node. Additionally, a scalar is used to represent the action $a \in [n]$.

**Embeddings for the P MDP.** The state of the $n$ dimensional P MDP is denoted as $s = (\mathbf{x}, \mathbf{c}, \mathbf{v})$. Assuming the upper bound of the size of the circuit is $P(n)$, similar to the previous CVP MDP, a $3P(n)$-dimension vector $\mathbf{e}_\mathbf{c}$ is employed to represent the circuit $\mathbf{c}$. Meanwhile, a $P(n)$ dimensional vector $\mathbf{v}$ is used to represent the value vector of the circuit, and a $n$ dimensional vector $\mathbf{x}$ represents the input string. In this representation, an integer is used to denote the character of the input, the index of the circuit, the value of a node, or the type of a node. Therefore, a $4P(n) + n$ dimensional vector $\mathbf{e}_s = (\mathbf{x}, \mathbf{e}_\mathbf{c}, \mathbf{v})$ is used to represent the state $s = (\mathbf{x}, \mathbf{c}, \mathbf{v})$. Additionally, a scalar is used to represent the action $a \in [P(n)]$.

## G.2 Proof of Theorem 5.2

*Proof of Theorem 5.2.* We show that the model, encompassing both the reward and transition kernel, of both 3-SAT MDPs and NP MDPs can be represented by MLP with constant layers for each respective case.

**Reward Function of 3-SAT MDP.** First of all, we will prove that the reward function of 3-SAT MDP can be implemented by a constant layer MLP. Denoting the input

$$\mathbf{e}_0 = (\mathbf{e}_s, a) = (\mathbf{e}_\psi, \mathbf{v}, k, a),$$

which embeds the state $s = (\psi, \mathbf{v}, k)$ and the action $a$. For the clarity of presentation, we divide the MLP into three modules and demonstrate each module in detail.

**Module 1.** The first module is designed to substitute the variable in $\psi$. Let $\psi'$ be the formula obtained from $\psi$ by substituting all variables and containing only Boolean value. Recall that we use $[2n]$ to represent $\{u_1, \neg u_1, \cdots, u_n, \neg u_n\}$. We define a $2n$ dimensional vector $\widetilde{\mathbf{v}}$ such that, for any literal $\tau \in \{u_1, \neg u_1, \cdots, u_n, \neg u_n\}$, $\widetilde{\mathbf{v}}[\tau] = \mathbf{v}[i]$ if $\tau = u_i$ and $\widetilde{\mathbf{v}}[\tau] = \neg\mathbf{v}[i]$ if $\tau = \neg u_i$. Hence, given a literal $\tau \in [2n]$, we can get its value $\widetilde{\mathbf{v}}[\tau]$ by

$$\widetilde{\mathbf{v}}[\tau] = \text{ReLU}\Big( \sum_{i \in [n]} \text{ReLU}(\mathbf{v}[i] + \mathbb{1}[\tau = u_i] - 1) + \sum_{i \in [n]} \text{ReLU}(\mathbb{1}[\tau = \neg u_i]) - \mathbf{v}[i] \Big).$$

According to Lemma I.5, the function $\mathbb{1}[\tau = u_i]$ and $\mathbb{1}[\tau = \neg u_i]$ can be implemented by the constant layer MLP of polynomial size. So we can get the output of Module 1, which is

$$\mathbf{e}_1 = (\mathbf{e}_{\psi'}, \mathbf{v}, k, a).$$

**Module 2.** The next module is designed to compute the value of $\psi'$. Given the value of each literal $\alpha_{i,j}$ we can compute the value of $\psi'$, i.e., $\psi(\mathbf{v})$, by the following formula:

$$\psi(\mathbf{v}) = \text{ReLU}\Big( \sum_{i \in [n]} \Big( \text{ReLU}(\alpha_{i,1} + \alpha_{i,2} + \alpha_{i,3}) - \text{ReLU}(\alpha_{i,1} + \alpha_{i,2} + \alpha_{i,3} - 1) \Big) - n + 1 \Big).$$

So we can get the output of Module 2, which is

$$\mathbf{e}_2 = (\psi(\mathbf{v}), k, a).$$

**Module 3.** The last module is designed to compute the final output $r(s, a)$. Given the input $\mathbf{e}_2 = (\psi(\mathbf{v}), k, a)$, we can compute the output $r(s, a)$ by

$$r(s, a) = \mathbb{1}[\psi(\mathbf{v}) \wedge (k = n + 1)] + 0.5 \cdot \mathbb{1}[k = 2n + 2].$$

While the input can take on polynomial types of values at most, according to Lemma I.5, we can use the MLP to implement this function. Therefore, we can use a constant layer MLP with polynomial hidden dimension to implement the reward function of $n$ dimension 3-SAT MDP.

**Transition Kernel of 3-SAT MDP.** We can use a constant layer MLP with polynomial hidden dimension to implement the transition kernel of 3-SAT MDP. Denote the input

$$\mathbf{e}_0 = (\mathbf{e}_s, a) = (\mathbf{e}_\psi, \mathbf{v}, k, a),$$

which embeds the state $s = (\psi, \mathbf{v}, k)$ and the action $a$. We only need to modify the embeddings $\mathbf{v}$ and $k$ and denote them as $\mathbf{v}'$ and $k'$. According to (E.1) in the proof of the transition kernel of Theorem 3.3, we have the output $\mathbf{v}'$ and $k$ of the following form:

$$\mathbf{e}'_\psi = \mathbf{e}_\psi, \qquad \mathbf{v}'[i] = (\mathbf{v}[i] \wedge \mathbb{1}[i \neq k]) \vee (a \wedge \mathbb{1}[i = k]),$$
$$k' = (k + 1) \cdot \mathbb{1}[k \geq 1] + \mathbb{1}[k = 0 \wedge a = 1] + (n + 2) \cdot \mathbb{1}[a = k = 0]. \tag{G.1}$$

In (G.1), each output element is determined by either an element or a tuple of elements that have polynomial value types at most. Therefore, according to Lemma I.5, each output element can be computed by constant-layer MLP of polynomial size, and the overall output can be represented by a constant layer MLP of polynomial size.

**Reward Function of** NP **MDP.** We can use a constant layer MLP with polynomial hidden dimension to implement the reward function of NP MDP. Denote the input

$$\mathbf{e}_0 = (\mathbf{e}_s, a) = (\mathbf{e}_c, k, a),$$

which embeds the state $s = (c, k) = (s_M, \mathbf{t}, l, k)$ and the action $a$. By (E.2) in the proof of Theorem 3.8, the transition function can be computed by the formula

$$r(s, a) = \mathbb{1}[(s_M = s_{\text{accept}}) \wedge (k = P(n) + 1)] + 0.5 \cdot \mathbb{1}[k = 2P(n) + 2].$$

Therefore, the reward $r(s, a)$ is determined by a tuple $(s_M, a, k)$, which has polynomial value types at most. According to Lemma I.5, we can compute this function by a constant-layer MLP of polynomial hidden dimension.

**Transition Kernel of** NP **MDP.** Finally, we switch to the transition kernel of NP MDP. Denote the input

$$\mathbf{e}_0 = (\mathbf{e}_s, a) = (\mathbf{e}_c, k, a),$$

which embeds the state $s = (c, k) = (s_M, \mathbf{t}, l, k)$ and the action $a$ and denote the dimension of $\mathbf{t}$ is $P(n)$. We obtain the final output in three steps. The first step is to extract the content $\chi$ of the location $l$ on the tape by the formula

$$\chi = \bigvee_{i \in [P(n)]} (\mathbf{t}[i] \wedge \mathbb{1}[i = l]).$$

We further convert it to the form in MLP by

$$\chi = \text{ReLU}\Big( \sum_{i \in [P(n)]} \chi_i' \Big) - \text{ReLU}\Big( \sum_{i \in [P(n)]} \chi_i' - 1 \Big), \quad \text{where } \chi_i' = \mathbf{t}[i] \cdot \mathbb{1}[i = l].$$

Regarding the configuration $c' = (s_M', \mathbf{t}', l')$ defined in (3.4), we notice that

  (i) the Turing Machine state $s_M'$ is determined by $s, a$ and $\chi$;

  (ii) the content of the location $l$ on the tape, $\mathbf{t}'[l]$, is determined by $s_M$, $a$ and $\chi$, whereas the contents of the other locations on the tape remain unaltered, i.e., $t'[i] = t[i]$ for $i \neq k$;

  (iii) the pointer $l'$ is determined by $l, s_M, a$ and $\chi$.

In addition, the number of steps $k'$ is determined by $k$ and $a$. Therefore, each output element is determined by an element or a tuple of elements having polynomial value types at most. According to Lemma I.5, a constant-layer MLP of polynomial size can compute each output element, and the overall output can be computed by a constant-layer MLP of polynomial size. □

### G.3 Proof of Theorem 5.3

*Proof of Theorem 5.3.* According to Lemma I.7, the computational complexity of MLP with constant layer, polynomial hidden dimension (in $n$), and ReLU as the activation function is upper-bounded by $\text{TC}^0$. On the other hand, according to Theorem 3.3 and Theorem 3.8, the computation of the optimal policy and optimal value function for the 3-SAT MDP and NP MDP is NP-complete. Therefore, the theorem holds under the assumption of $\text{TC}^0 \neq \text{NP}$. □

### G.4 Proof of Theorem 5.4

*Proof of Theorem 5.4.* We show that the reward function, transition kernel, and optimal policy of both CVP MDPs and P MDPs can be individually represented by MLP with constant layers and polynomial hidden dimension.

**Reward Function of CVP MDP.** We can use a constant layer MLP with polynomial hidden dimension to implement the reward function of CVP MDP. Denote the input

$$\mathbf{e}_0 = (\mathbf{e}_s, a) = (\mathbf{e}_c, \mathbf{v}, a),$$

which embeds the state $s = (\mathbf{c}, \mathbf{v})$ and the action $a$. According to the definition, we can represent the reward as

$$r(s, a) = \mathbb{1}[\mathbf{v}[n] = 1].$$

By Lemma I.5, we know that $\mathbb{1}[\mathbf{v}[n] = 1]$ can be represented by a constant-layer MLP with polynomial hidden dimension. Hence, the reward function of CVP MDP can be represented by a constant-layer MLP with polynomial hidden dimension.

**Transition Kernel of CVP MDP.** We can use a constant layer MLP with polynomial hidden dimension to implement the transition kernel of CVP MDP. Denote the input

$$\mathbf{e}_0 = (\mathbf{e}_s, a) = (\mathbf{e_c}, \mathbf{v}, a),$$

which embeds the state $s = (\mathbf{c}, \mathbf{v})$ and the action $a$. Given an index $i$, we can fetch the node $i$ and its value by

$$
\begin{aligned}
\mathbf{v}[i] &= \sum_{j \in [n]} \alpha_{i,j}, \quad \text{where } \alpha_{i,j} = \mathbb{1}[i = j] \cdot \mathbf{v}[i], \\
\mathbf{c}[i] &= \sum_{j \in [n]} \beta_{i,j}, \quad \text{where } \beta_{i,j} = \mathbb{1}[i = j] \cdot \mathbf{c}[i].
\end{aligned}
\tag{G.2}
$$

Then, compute the output of node $i$ and denote it as $\mathbf{o}[i]$. $\mathbf{o}[i]$ is determined by the $i$-th node $c_i$ and its input, therefore, can be computed by a constant layer MLP with polynomial hidden dimension according to Lemma I.5. Denoting the output as $(\mathbf{e}'_{\mathbf{c}}, \mathbf{v}')$, according to the definition of the transition kernel, we have

$$\mathbf{e}'_{\mathbf{c}} = \mathbf{e}, \qquad \mathbf{v}[i]' = \mathbf{v}[i] \cdot \mathbb{1}[i \neq a] + \mathbf{v}[i] \cdot \mathbb{1}[i = a]$$

Therefore, we can compute the transition kernel of CVP MDP by a constant layer MLP with polynomial hidden dimension.

**Optimal Policy of CVP MDP.** We prove that the MLP can implement the optimal policy, which is specified in the proof of Theorem 4.2 (Appendix F.1). For the convenience of reading, we present the optimal policy here again. Given a state $s = (\mathbf{c}, \mathbf{v})$, denoting $\mathbf{c}[i] = (\mathbf{c}[i][1], \mathbf{c}[i][2], g_i)$, we define $\mathcal{G}(s)$

$$
\begin{aligned}
\mathcal{G}(s) = &\{i \in [n] \mid g_i \in \{\wedge, \vee\}, \mathbf{v}[\mathbf{c}[i][1]], \mathbf{v}[\mathbf{c}[i][2]] \in \{0, 1\}, \mathbf{v}[i] = \texttt{Unknown}\} \\
&\cup \{i \in [n] \mid g_i \in \{\neg\}, \mathbf{v}[\mathbf{c}[i][1]] \in \{0, 1\}, \mathbf{v}[i] = \texttt{Unknown}\} \\
&\cup \{i \in [n] \mid g_i \in \{0, 1\}, \mathbf{v}[i] = \texttt{Unknown}\}.
\end{aligned}
\tag{G.3}
$$

The set $\mathcal{G}(s)$ defined in (G.3) denotes the indices for which inputs have been computed, and the output has not been computed. Consequently, the optimal policy $\pi^*(s)$ is expressed as $\pi^*(s) = \min \mathcal{G}(s)$. Subsequently, we aim to demonstrate that a constant-layer MLP with polynomial hidden dimension can compute the optimal policy $\pi^*$. Let $\Upsilon[i]$ denote the indicator of whether $i \in \mathcal{G}(s)$, i.e., $\Upsilon[i] = 1$ signifies that $i \in \mathcal{G}(s)$, while $\Upsilon[i] = 0$ denotes that $i \notin \mathcal{G}(s)$. According to (G.3), we can compute $\Upsilon[i]$ depends on the $i$-th node $\mathbf{c}[i]$, its inputs and output, therefore, can be computed by a constant-layer MLP with polynomial hidden dimension. Then we can express the optimal policy $\pi^*$ as:

$$\pi^*(s) = \text{ReLU}\Big( \sum_{i \in [n]} \Upsilon'[i] \Big), \qquad \text{where } \Upsilon'[i] = \text{ReLU}\Big( 1 - \sum_{j < i} \Upsilon[j] \Big).$$

Therefore, we can compute the optimal policy by a constant-layer MLP with polynomial size.

**Reward Function of P MDP.** Denote the input

$$\mathbf{e}_0 = (\mathbf{e}_s, a) = ((\mathbf{x}, \mathbf{e_c}, \mathbf{v}), a),$$

which embeds the state $s = (\mathbf{x}, c, \mathbf{v})$ and the action $a$ and denote the size of the circuit $c$ as $P(n)$. According to the definition, we have

$$r(s, a) = \mathbb{1}[\mathbf{v}[P(n)] = 1].$$

Owning to Lemma I.5, we can compute the reward function of P MDP by a constant layer MLP with polynomial hidden dimension.

**Transition Kernel of** P **MDP.** Denote the input

$$\mathbf{e}_0 = (\mathbf{e}_s, a) = ((\mathbf{x}, \mathbf{e_c}, \mathbf{v}), a),$$

which embeds the state $s = (\mathbf{x}, \mathbf{c}, \mathbf{v})$ and the action $a$. We can fetch the node $i$ and its value by (G.2). Then we compute the output of node $i$ and denote it as $\mathbf{o}[i]$, where $\mathbf{o}[i]$ is determined by the $i$-th node $\mathbf{c}[i]$ and its inputs. Hence, by Lemma I.5, $\mathbf{o}[i]$ can be computed by a constant-layer MLP with polynomial hidden dimension. Denoting the output as $(\mathbf{x}', \mathbf{e_c'}, \mathbf{v}')$, together wths the definition of the transition kernel, we have

$$\mathbf{x}' = \mathbf{x}, \qquad \mathbf{e_c'} = \mathbf{e_c}, \qquad \mathbf{v}'[i] = \mathbf{v}[i] \cdot \mathbb{1}[i \neq a] + \mathbf{v}[i] \cdot \mathbb{1}[i = a].$$

Therefore, we can compute the transition kernel of P MDP by a constant layer MLP with polynomial hidden dimension.

**Optimal Policy of** P **MDP.** We prove that the MLP can efficiently implement the optimal policy in the proof of Theorem 4.5 (Appendix F.2). For completeness, we present the definition of optimal policy here. Given a state $s = (\mathbf{x}, \mathbf{c}, \mathbf{v})$, denoting $\mathbf{c}[i] = (\mathbf{c}[i][1], \mathbf{c}[i][2], g_i)$, let $\widetilde{\mathcal{G}}(s)$ be a set defined by:

$$
\begin{aligned}
\widetilde{\mathcal{G}}(s) = & \{i \in [P(n)] \mid g_i \in \{\wedge, \vee\}, \mathbf{v}[\mathbf{c}[i][1]], \mathbf{v}[\mathbf{c}[i][2]] \in \{0, 1\}, \mathbf{v}[i] = \texttt{Unknown}\} \\
& \cup \{i \in [P(n)] \mid g_i = \neg, \mathbf{v}[\mathbf{c}[i][1]] \in \{0, 1\}, \mathbf{v}[i] = \texttt{Unknown}\} \\
& \cup \{i \in [P(n)] \mid g_i = \texttt{Input}, \mathbf{v}[i] = \texttt{Unknown}\}.
\end{aligned}
\tag{G.4}
$$

The set $\widetilde{\mathcal{G}}(s)$ defined in (G.4) denotes the indices for which inputs have been computed, and the output has not been computed. With this set, the optimal policy $\pi^*(s)$ is expressed as $\pi^*(s) = \min \widetilde{\mathcal{G}}(s)$. Subsequently, we aim to demonstrate that a constant-layer MLP with polynomial hidden dimension can compute the optimal policy $\pi^*$. Let $\widetilde{\Upsilon}[i]$ denote the indicator of whether $i \in \widetilde{\mathcal{G}}(s)$, i.e., $\widetilde{\Upsilon}[i] = 1$ signifies that $i \in \widetilde{\mathcal{G}}(s)$, while $\widetilde{\Upsilon}[i] = 0$ denotes that $i \notin \widetilde{\mathcal{G}}(s)$. Using (G.4), the computation of $\widetilde{\Upsilon}[i]$ depends on the $i$-th node $c_i$, its inputs, and output. This observation, together with Lemma I.5, allows for the computation through a constant-layer MLP with polynomial hidden dimension. Employing this notation, we can formulate the following MLP expression for the optimal policy $\pi^*$:

$$\pi^*(s) = \mathrm{ReLU}\Big( \sum_{i \in [P(n)]} \widetilde{\Upsilon}'[i] \Big), \qquad \text{where } \widetilde{\Upsilon}'[i] = \mathrm{ReLU}\Big( 1 - \sum_{j < i} \widetilde{\Upsilon}[j] \Big).$$

Therefore, the optimal policy of P MDP can be represented by a constant-layer MLP with polynomial hidden dimension. $\qquad \square$

### G.5   Proof of Theorem 5.5

*Proof of Theorem 5.5.* By Lemma I.7, we have that the computational complexity of MLP with constant layer, polynomial hidden dimension (in $n$), and ReLU as the activation function is upper-bounded by $\mathsf{TC}^0$. On the other hand, by Theorem 4.2 and 4.5, the computation of the optimal value Function of CVP MDP and P MDP is P-complete. Therefore, we conclude the proof under the assumption of $\mathsf{TC}^0 \neq \mathsf{P}$. $\qquad \square$

## H   Discussions on the Extensions of Our Results

### H.1   Connections to POMDPs

For the partially observable Markov decision process (POMDP), the agent can only receive the observation $o \in \mathcal{O}$, generated by the emission function $\mathbb{O} = \{\mathbb{O}_h : \mathcal{S} \mapsto \Delta(\mathcal{O})\}_{h \in [H]}$. one can choose the observation as a substring of the state, and the representation complexity of the emission function remains low, similar to the transition kernel. However, the optimal policy and optimal value function may depend on the full history rather than just the current state, leading to a higher representation complexity for these two quantities. Consequently, in the presence of partial observations, the representation complexity gap between model-based RL and model-free RL could be more pronounced.

## H.2 Extension to Stochastic MDPs

In this section, we demonstrate how our construction of the 3-SAT MDP, NP MDP, CVP MDP, and P MDP can be seamlessly extended to their stochastic counterparts. We offer a detailed extension of the 3-SAT MDP to its stochastic version and outline the conceptual approach for extending other types of MDPs to their stochastic counterparts.

**Stochastic 3-SAT MDP.** First, we add some randomness to the transition kernel and reward function. Moreover, to maintain the properties in Theorem 3.3, Theorem 5.3, and Theorem 5.2 of 3-SAT MDP, we slightly modify the action space and extend the planning horizon.

**Definition H.1** (Stochastic 3-SAT MDP). For any $n \in \mathbb{N}_+$, let $\mathcal{V} = \{u_1, \neg u_1, \cdots, u_n, \neg u_n\}$ be the set of literals. An $n$-dimensional stochastic 3-SAT MDP $(\mathcal{S}, \mathcal{A}, H, \mathcal{P}, r)$ is defined as follows. The state space $\mathcal{S}$ is defined by $\mathcal{S} = \mathcal{V}^{3n} \times \{0, 1, \texttt{Next}\}^n \times (\{0\} \cup [n+2])$, where each state $s$ can be denoted as $s = (\psi, \mathbf{v}, k)$. In this representation, $\psi$ is a 3-CNF formula consisting of $n$ clauses and represented by its $3n$ literals, $\mathbf{v} \in \{0, 1\}^n$ can be viewed as an assignment of the $n$ variables and $k$ is an integer recording the number of actions performed. The action space is $\mathcal{A} = \{0, 1\}$ and the planning horizon is $H = n^2 + n + 2$. Given a state $s = (\psi, \mathbf{v}, k)$, for any $a \in \mathcal{A}$, the reward $r(s, a)$ is defined by:

$$r(s, a) = \begin{cases} 1 & \text{If } \mathbf{v} \text{ is a satisfiable assignment of } \psi, k = n+1 \text{ and } a = \texttt{Next}, \\ \frac{1}{2} & \text{If } k = n^2 + 2n + 2, \\ 0 & \text{Otherwise.} \end{cases} \tag{H.1}$$

Moreover, the transition kernel is stochastic and takes the following form:

$$\mathcal{P}\big((\psi, \mathbf{v}, k), a\big) = \begin{cases} (\psi, \mathbf{v}, n+2) & \text{If } a = k = 0, \\ (\psi, \mathbf{v}, 1) & \text{If } a = 1 \text{ and } k = 0, \\ (\psi, \mathbf{v}, k+1) & \text{If } a = \texttt{Next}, \\ (\psi, \mathbf{v}', k) & \text{If } k \in [n] \text{ and } a \in \{0, 1\} \\ (\psi, \mathbf{v}, k) & \text{If } k > n \text{ and } a \in \{0, 1\}, \end{cases} \tag{H.2}$$

where $\mathbf{v}'$ is obtained from $\mathbf{v}$ by setting the $k$-th bit as $a$ with probability $\frac{2}{3}$ and as $1-a$ with probability $\frac{1}{3}$, and leaving other bits unchanged, i.e.,

$$\mathbf{v}'[k] = \begin{cases} a & \text{with probability } \frac{2}{3} \\ 1-a & \text{with probability } \frac{1}{3} \end{cases} \qquad \mathbf{v}'[k'] = \mathbf{v}[k'] \text{ for } k' \neq k.$$

Given a 3-CNF formula $\psi$, the initial state of the 3-SAT MDP is $(\psi, \mathbf{0}_n, 0)$.

**Theorem H.2** (Representation complexity of Stochastic 3-SAT MDP). Let $\mathcal{M}_n$ be the $n$-dimensional stochastic 3-SAT MDP in Definition H.1. The transition kernel $\mathcal{P}$ and the reward function $r$ of $\mathcal{M}_n$ can be computed by circuits with polynomial size (in $n$) and constant depth, falling within the circuit complexity class $\mathsf{AC}^0$. However, computing the optimal value function $Q_1^*$ and the optimal policy $\pi^*$ of $\mathcal{M}_n$ are both NP-hard under the polynomial time reduction.

*Proof of Theorem H.2.* We investigate the representation complexity of the reward function, transition kernel, optimal value function, and optimal policy in sequence.

**Reward Function.** The reward function is the same as the deterministic version, therefore, we can apply the proof of Theorem 3.3 and conclude that the complexity of the reward function falls in $\mathsf{AC}^0$.

**Transition Kernel.** Then, we will implement the transition kernel by $\mathsf{AC}^0$ circuits. Slightly different from the deterministic version, in this case, the input of the transition kernel $\mathcal{P}$ are two states $s = (\psi, \mathbf{v}, k), s' = (\psi', \mathbf{v}', k')$ and action $a$, and the output is the probability of transition from $s$ to $s'$. Given the input $\mathbf{v}, k, \mathbf{v}', k'$ and $a$, we have the output as follows:

- The output is 1 if the input satisfies the following four equations:

$$a = \texttt{Next}, \qquad \mathbf{v} = \mathbf{v}', \qquad \psi = \psi',$$
$$k' = (k+1) \cdot \mathbb{1}[k \geq 1] + \mathbb{1}[k = 0 \wedge a = 1] + (n+2) \cdot \mathbb{1}[a = k = 0].$$

- The output is $\frac{2}{3}$ if the input satisfies the following four equations:

$$a \in \{0,1\}, \qquad k = k', \qquad \psi = \psi', \qquad \mathbf{v}'[i] = (\mathbf{v}[i] \wedge \mathbb{1}[i \neq k]) \vee (a \wedge \mathbb{1}[i = k]).$$

- The output is $\frac{1}{3}$ if the input satisfies the following four equations:

$$a \in \{0,1\}, \qquad k = k', \qquad \psi = \psi', \qquad \mathbf{v}'[i] = \neg(\mathbf{v}[i] \wedge \mathbb{1}[i \neq k]) \vee (a \wedge \mathbb{1}[i = k]).$$

- The output is $0$ otherwise.

It is noted that each element in the previous equations is determined by at most $O(\log n)$ bits. Therefore, according to Lemma I.4, the condition judgments can be computed by two-layer circuits of polynomial size, and the overall output can be computed by $\mathsf{AC}^0$ circuits.

**Optimal Value Function.** Next, we will prove the NP-hardness of computing the optimal value function. Similar to the optimal value function part of the proof of Theorem 3.3, we formulate a simpler decision version of this problem: given a state $s = (\psi, \mathbf{v}, k)$, an action $a$ and a number $\gamma$ as input, and the goal is to determine whether $Q_1^*(s, a) > \frac{1}{2}$. According to the well-known Cook-Levin theorem (Lemma I.1), the 3-SAT problem is NP-complete. Thus, our objective is to provide a polynomial time reduction from the 3-SAT problem to the computation of the optimal value function for the 3-SAT MDP. Given a Boolean formula of length $n$, the number of variables is at most $n$. Then, we can pad several meaningless clauses such as $(u_1 \vee \neg u_1 \vee u_1)$ to obtain the 3-CNF Boolean formula $\psi'$ with $n$ clauses. When the Boolean formula $\psi$ is not satisfiable, the value of $Q_1^*((\psi', \mathbf{0}_n, 0), 1)$ is $0$. When the Boolean formula $\psi$ is satisfiable, we will prove that the probability of the reward $1$ is higher than $\frac{1}{2}$. Note that, when $\psi$ is satisfiable, we only need to modify the values of the variables at most $n$ times to get a satisfiable assignment in the deterministic case. In the stochastic case, we have $n^2$ chances to modify the value of a variable with a success probability of $\frac{2}{3}$, and to get the reward, we only need $n$ times success. Therefore, we can compute the probability of getting the reward as

$$\Pr(n \text{ times success in } n^2 \text{ chances}) \geq \Big( \Pr(\text{one success in } n \text{ chances}) \Big)^n = \Big(1 - \frac{1}{3^n}\Big)^n$$
$$> \Big(1 - \frac{1}{2n}\Big)^n \geq \frac{1}{2}. \tag{H.3}$$

Therefore, $\psi$ is satisfiable if and only if $Q_1^*((\psi', \mathbf{0}_n, 0), 1) > \frac{1}{2}$, we can conclude that computing the optimal value function is NP-hard.

**Optimal Policy.** Finally, we will prove that the problem of computing the optimal policy is NP-hard. According to the well-known Cook-Levin theorem (Lemma I.1), the 3-SAT problem is NP-complete. Thus, our objective is to provide a polynomial time reduction from the 3-SAT problem to the problem of computing the optimal policy of 3-SAT MDP. Given a Boolean formula of length $n$, the number of variables is at most $n$. Then, we can pad several meaningless clauses such as $(u_1 \vee \neg u_1 \vee u_1)$ to obtain the 3-CNF Boolean formula $\psi'$ with $n$ clauses. When the Boolean formula $\psi$ is satisfiable, according to (H.3), we have

$$Q_1^*((\psi', \mathbf{0}_n, 0), 1) > 0.7 > Q_1^*((\psi', \mathbf{0}_n, 0), 0),$$

which gives that $\pi^*(\psi', \mathbf{0}_n, 0) = 1$. When the Boolean formula $\psi$ is not satisfiable, we have

$$Q_1^*((\psi', \mathbf{0}_n, 0), 1) = 0 < Q_1^*((\psi', \mathbf{0}_n, 0), 0),$$

which implies that $\pi^*(\psi', \mathbf{0}_n, 0) = 0$. So the Boolean formula $\psi$ is satisfiable if and only if $\pi^*(\psi', \mathbf{0}_n, 0) = 1$, which concludes that the problem of computing the optimal policy is NP-hard. $\square$

Almost the same as the stochastic version of 3-SAT MDP, we can construct the stochastic version NP MDP. And under the assumption of $\mathcal{L} \in \mathsf{NP}$, the same theorem as Theorem H.2 will hold. More exactly, the complexity of the transition kernel and the reward function of the MDP based on $\mathcal{L}$ fall in $\mathsf{AC}^0$, and the complexity of the optimal policy and optimal value function are NP-hard. Moreover, similar to the case of the stochastic version of 3-SAT MDP, we add some randomness to the transition function of CVP MDP and P MDP. In the deterministic version of the transition function, given the action $i$, we will compute the value of the $i$-th node. In contrast, the computation of the value of $i$-th node will be correct with the probability of $\frac{2}{3}$ and will be incorrect with the probability of $\frac{1}{3}$.

And we extend the planning horizon to $O(n^2)$ or $O(P(n)^2)$. Then we can get similar conclusions as Theorems 4.2 and 4.5. To avoid repetition, we only provide the construction and corresponding theorem of stochastic CVP and omit the proof and the detailed extension to stochastic P MDPs.

**Definition H.3** (Stochastic CVP MDP). An $n$-dimensional Stochastic CVP MDP is defined as follows. Let $\mathcal{C}$ be the set of all circuits of size $n$. The state space $\mathcal{S}$ is defined by $\mathcal{S} = \mathcal{C} \times \{0, 1, \texttt{Unknown}\}^n$, where each state $s$ can be represented as $s = (\mathbf{c}, \mathbf{v})$. Here, $\mathbf{c}$ is a circuit consisting of $n$ nodes with $\mathbf{c}[i] = (\mathbf{c}[i][1], \mathbf{c}[i][2], g_i)$ describing the $i$-th node, where $\mathbf{c}[i][1]$ and $\mathbf{c}[i][2]$ indicate the input node and $g_i$ denotes the type of gate (including $\wedge, \vee, \neg, 0, 1$). When $g_i \in \{\wedge, \vee\}$, the outputs of $\mathbf{c}[i][1]$-th node and $\mathbf{c}[i][2]$-th node serve as the inputs; and when $g_i = \neg$, the output of $\mathbf{c}[i][1]$-th node serves as the input and $\mathbf{c}[i][2]$ is meaningless. Moreover, the node type of 0 or 1 denotes that the corresponding node is a leaf node with a value of 0 or 1, respectively, and therefore, $\mathbf{c}[i][1], \mathbf{c}[i][2]$ are both meaningless. The vector $\mathbf{v} \in \{0, 1, \texttt{Unknown}\}^n$ represents the value of the $n$ nodes, where the value $\texttt{Unknown}$ indicates that the value of this node has not been computed and is presently unknown. The action space is $\mathcal{A} = [n]$ and the planning horizon is $H = n + 1$. Given a state-action pair $(s = (\mathbf{c}, \mathbf{v}), a)$, its reward $r(s, a)$ is given by:

$$r(s, a) = \begin{cases} 1 & \text{If } \mathbf{v} \text{ contains correct value of the } n \text{ gates and the value of the output gate } \mathbf{v}[n] = 1, \\ 0 & \text{Otherwise.} \end{cases}$$

Moreover, the transition kernel is deterministic and can be defined as follows:

$$\mathcal{P}\big((\mathbf{c}, \mathbf{v}), a\big) = (\mathbf{c}, \mathbf{v}').$$

Here, $\mathbf{v}'$ is obtained from $\mathbf{v}$ by computing and substituting the value of node $a$ with the probability of $\frac{2}{3}$. More exactly, if the inputs of node $a$ have been computed, we can compute the output of the node $a$ and denote it as $\mathbf{o}[a]$. Let $\widetilde{\mathbf{o}}[a]$ be a random variable getting value of $\mathbf{o}[a]$ with probability of $\frac{2}{3}$ and getting value of $\texttt{Unknown}$ with probability of $\frac{1}{3}$. Then we have

$$\mathbf{v}'[j] = \begin{cases} \mathbf{v}[j] & \text{If } a \neq j, \\ \widetilde{\mathbf{o}}[a] & \text{If } a = j \text{ and the inputs of node } a \text{ have been computed}, \\ \texttt{Unknown} & \text{If } a = j \text{ and the inputs of node } a \text{ have not been computed.} \end{cases}$$

Given a circuit $\mathbf{c}$, the initial state of CVP MDP is $(\mathbf{c}, \mathbf{v}_{\text{unknown}})$ where $\mathbf{v}_{\text{unknown}}$ denotes the vector containing $n$ Unknown values.

**Theorem H.4** (Representation complexity of Stochastic CVP MDP). Let $\mathcal{M}_n$ be the $n$-dimensional stochastic CVP MDP in Definition H.1. The transition kernel $\mathcal{P}$, the reward function $r$, and the optimal policy $\pi^*$ of $\mathcal{M}_n$ can be computed by circuits with polynomial size (in $n$) and constant depth, falling within the circuit complexity class $\mathsf{AC}^0$. However, computing the optimal value function $Q_1^*$ of $\mathcal{M}_n$ are P-hard under the log space reduction.

# I   Auxiliary Lemmas

**Lemma I.1** (Cook-Levin Theorem [12, 31]). The 3-SAT problem is NP-complete.

**Lemma I.2** (P-completeness of CVP [28]). The CVP is P-complete.

**Lemma I.3** (Theorem 6.5 in [3]). A language has log-space-uniform circuits of polynomial size if and only if it is in P.

**Lemma I.4** (Implement Any Boolean Function). For every Boolean function $f : \{0, 1\}^l \to \{0, 1\}$, there exists a two layer circuit $\mathbf{c}$ of size $\mathcal{O}(l \cdot 2^l)$ such that $\mathbf{c}(\mathbf{u}) = f(\mathbf{u})$ for all $\mathbf{u} \in \{0, 1\}^l$.

*Proof.* For every $\mathbf{v} \in \{0, 1\}^l$, let

$$\mathbf{c}_{\mathbf{v}}(\mathbf{u}) = \bigwedge_{i \in [n]} g_{\mathbf{v}[i]}(\mathbf{u}_i),$$

where $g_0(\mathbf{u}[i]) = \neg \mathbf{u}[i]$ and $g_1(\mathbf{u}[i]) = \mathbf{u}[i]$. Then we have $\mathbf{c}_{\mathbf{v}}(\mathbf{u}) = 1$ if and only if $\mathbf{u} = \mathbf{v}$. When there exists $\mathbf{u} \in \{0, 1\}$ such that $f(\mathbf{u}) = 1$, we can construct the two-layer circuit as

$$\mathbf{c}(\mathbf{u}) = \bigvee_{\mathbf{v} \in \{\mathbf{v} | f(\mathbf{v}) = 1\}} \big(\mathbf{c}_{\mathbf{v}}(\mathbf{u})\big).$$

By definition, we can verify that $f(\mathbf{u}) = \mathbf{c}(\mathbf{u})$ for any $\mathbf{u} \in \{0,1\}^l$. When $f(\mathbf{u}) = 0$ for all $\mathbf{u} \in \{0,1\}^l$, we can construct the circuit as $\mathbf{c}(\mathbf{u}) = 0$. Therefore, for every Boolean function $f : \{0,1\}^l \to \{0,1\}$, there exists a two-layer circuit $\mathbf{c}$ of size $\mathcal{O}(l \cdot 2^l)$ such that $\mathbf{c}(\mathbf{u}) = f(\mathbf{u})$ for all $\mathbf{u} \in \{0,1\}^l$, which concludes the proof of Lemma I.4. □

**Lemma I.5** (Looking-up Table for MLP). For any function $f : \mathcal{X} \to \mathbb{R}$, where $\mathcal{X}$ is a finite subset of $\mathbb{R}^l$, there exists a constant-layer MLP $f_{\mathrm{MLP}}$ with hidden dimension $\mathcal{O}(l \cdot |\mathcal{X}|)$ such that $f_M(\mathbf{x}) = f(\mathbf{x})$ for all $\mathbf{x} \in \mathcal{X}$.

*Proof.* Denote the minimum gap between each pair of elements in $\mathcal{X}$ by $\delta_{\min}$, i.e.,

$$\delta_{\min} = \min_{\mathbf{u},\mathbf{v} \in \mathcal{X}} \|\mathbf{u} - \mathbf{v}\|_\infty.$$

For each $\mathbf{u} \in \mathcal{X}$, we can construct an MLP as

$$f_{\mathbf{u}}(\mathbf{x}) = \mathrm{ReLU}\Big( \sum_{i \in [l]} f_{\mathbf{u},i}(\mathbf{x}) - (l-1) \cdot \delta_{\min} \Big),$$

where

$$f_{\mathbf{u},i}(\mathbf{x}) = \mathrm{ReLU}\Big( 2\delta_{\min} - 2 \cdot \mathrm{ReLU}(\mathbf{x}[i] - \mathbf{u}[i]) - \mathrm{ReLU}(\mathbf{u}[i] - \mathbf{x}[i] + \delta_{\min}) \Big).$$

We have

$$f_{\mathbf{u}}(\mathbf{x}) = \begin{cases} \delta_{\min} & \text{If } \mathbf{x} = \mathbf{u}, \\ 0 & \text{If } \mathbf{x} \in \mathcal{X} \backslash \{\mathbf{u}\}. \end{cases}$$

Then we can construct the MLP $f_{\mathrm{MLP}}$ as

$$f_{\mathrm{MLP}}(\mathbf{x}) = \sum_{\mathbf{u} \in \mathcal{S}} \Big( \frac{f_{\mathbf{u}}(\mathbf{x})}{\delta_{\min}} \cdot f(\mathbf{u}) \Big).$$

It is easy to verify that $f_{\mathrm{MLP}}(\mathbf{x}) = f(\mathbf{x})$ for all $\mathbf{x} \in \mathcal{X}$, which concludes the proof of Lemma I.5. □

**Lemma I.6** (MLP can Implement Basic Gates in $\mathsf{TC}^0$). Given $\mathbf{x} \in \{0,1\}^n$ as input, constant-layer MLPs can implement the following basic operation functions:

- AND: $f_{\mathrm{AND}}(\mathbf{x}) = \bigwedge_{i \in [n]} \mathbf{x}[i]$;

- OR: $f_{\mathrm{OR}}(\mathbf{x}) = \bigvee_{i \in [n]} \mathbf{x}[i]$;

- NOT: $f_{\mathrm{NOT}}(\mathbf{x}[1]) = 1 - \mathbf{x}[1]$;

- Majority: $f_{\mathrm{MAJ}}(\mathbf{x}) = \mathbb{1}[\sum_{i \in [n]} \mathbf{x}[i] > \frac{n}{2}]$.

*Proof.* We express the four functions in the MLP forms as follows:

- $f_{\mathrm{AND}}(\mathbf{x}) = \mathrm{ReLU}(\sum_{i \in [n]} \mathbf{x}[i] - n + 1)$;

- $f_{\mathrm{OR}}(\mathbf{x}) = 1 - \mathrm{ReLU}(1 - \sum_{i \in [n]} \mathbf{x}[i])$;

- $f_{\mathrm{NOT}}(\mathbf{x}[1]) = 1 - \mathbf{x}[1]$;

- $f_{\mathrm{MAJ}}(\mathbf{x}) = \mathrm{ReLU}(2 \cdot \sum_{i \in [n]} \mathbf{x}[i] - n) - \mathrm{ReLU}(2 \cdot \sum_{i \in [n]} \mathbf{x}[i] - n - 1)$.

Therefore, the basic gates of $\mathsf{TC}^0$ can be implemented by constant-layer MLPs with ReLU as the activation function. □

**Lemma I.7** (Upper Bound of Expressive Power for MLP). Any log-precision MLP with constant layers, polynomial hidden dimension (in the input dimension), and ReLU as the activation function can be simulated by a L-uniform $\mathsf{TC}^0$ circuits.

*Proof.* In the previous work by [36], it was demonstrated that a Transformer with logarithmic precision, a fixed number of layers, and a polynomial hidden dimension can be simulated by a L-uniform $\mathsf{TC}^0$ circuit. The proof presented by [36] established the validity of this result specifically when the Transformer employs standard activation functions (e.g., ReLU, GeLU, and ELU) in the MLP. Considering that the MLP can be perceived as a submodule of the Transformer, it follows that a log-precision MLP with a consistent number of layers, polynomial hidden dimension, and ReLU as the activation function can be simulated by a L-uniform $\mathsf{TC}^0$ circuit. □

