# OpenReview forum: "Rethinking Model-based, Policy-based, and Value-based Reinforcement Learning via the Lens of Representation Complexity"
_NeurIPS.cc/2024/Conference — NeurIPS 2024 poster_

### Official Review · Reviewer_YjQh · 2024-07-09

**Soundness:** 3
**Presentation:** 3
**Contribution:** 2
**Rating:** 6
**Confidence:** 4

**Summary:**

The paper presents an analysis of the representation complexity, the necessary complexity of a circuit, between different paradigms in reinforcement learning. The authors show, with several reductions to well-known theoretical complexity classes, that MDPs exist in which representing a model is "easy" while representing a value function or policy is "hard".

**Strengths:**

The paper dives deep into an established conjecture in reinforcement learning, that learning a model of an environment can be easier than learning a value function. With a clever reduction to a simple exemplary class of MDPs, the authors show that is "difficulty" is measured by the complexity of a circuit necessary to represent the respective function, then classes of problems exist in which this is indeed true.
The mathematical derivations and proofs are straightforward and seem correct to the extent that I am able to verify them, although my expertise lies in RL and not complexity theory. They provide stronger results than previous work in the same direction [1] and seems like a good follow up work.

[1] ON REPRESENTATION COMPLEXITY OF MODEL-BASED AND MODEL-FREE REINFORCEMENT LEARNING, Hanlin Zhu, Baihe Huan, Stuart Russell, ICLR 2024

**Weaknesses:**

I have one major confusions which I list under questions since these are hopefully easily addressable in the rebuttal.

I think the problem of the "strictness" of the hierarchy should be very carefully addressed. It only weakens the conclusion, all the proofs are correct as far as I was able to verify, but I would encourage the authors to engage carefully. This is my main reason for recommending rejection: I think the works is interesting and meaningful, but the conclusion seems wrong due to the problem outlined below. I am very happy to discuss this point in the rebuttal and raise my scores accordingly if appropriate.

**Questions:**

The authors explicitly call the studied phenomenon a hierarchy, but I am uncertain if this is actually true. It seems intuitive that their should be MDPs in which the situation is reversed: representing the value function is easy, while representing the full model is difficult. This seems especially true if we don't talk about strict error free representation, but allow for errors. One intuitive reason for me to conjecture this is the existence of an arbitrarily complex model (i.e. one that transitions from a 3-SAT formula with a high likelihood to one of two states depending on the formula is satisfiable or not) but with 0 reward and therefore a trivial value function as well. I think this is my biggest problem with the paper so far: the construction implies the existence of one direction for the hierarchy, but does not imply that the other direction cannot exist.

This also makes the empirical section a bit more difficult: Why should we assume that the Mujoco locomotion environments fall into the category of problems described in the paper and not into the alternative? Empirically this seems to be true. Is there some reason to believe that "realistic" MDPs will exhibit this complexity relationship more, or is this an artifact of the choice of environments?

Other, minor issues:
- All figures are barely legible at a "reasonable" zoom level, I would encourage the authors to quickly redesign these.

**Limitations:**

See above

---

> ### Author Rebuttal · Authors · 2024-08-07
>
> Thank you for taking the time to review our paper. We appreciate your feedback and will address each of your concerns individually.
>
> ---
>
> **Q1:** The authors explicitly call the studied phenomenon a hierarchy, but I am uncertain if this is actually true. It seems intuitive that their should be MDPs in which the situation is reversed: representing the value function is easy, while representing the full model is difficult. This seems especially true if we don't talk about strict error free representation, but allow for errors. One intuitive reason for me to conjecture this is the existence of an arbitrarily complex model (i.e. one that transitions from a 3-SAT formula with a high likelihood to one of two states depending on the formula is satisfiable or not) but with 0 reward and therefore a trivial value function as well. I think this is my biggest problem with the paper so far: the construction implies the existence of one direction for the hierarchy, but does not imply that the other direction cannot exist.
>
>
> **A1:** Thank you for your question. We have acknowledged in our paper that our complexity hierarchy does not hold for all MDPs, and provide a detailed discussion in Appendix C.2. For the convenience of the reviewer, we paste Appendix C.2 below:
>
> > First, we wish to underscore that our identified representation complexity hierarchy holds in a general way. Theoretically, our proposed MDPs can encompass a wide range of problems, as any $\mathsf{NP}$ or $\mathsf{P}$ problems can be encoded within their structure. More crucially, our thorough experiments in diverse simulated settings support the representation complexity hierarchy we have uncovered. In fact, we have a generalized result establishing a hierarchy between policy-based RL and value-based RL, as stated in the following proposition:
>
> > **Proposition:** Given a Markov Decision Process (MDP) $\mathcal{M}=(\mathcal{S}, \mathcal{A}, H, \mathcal{P}, r)$, where $\mathcal{S}\subset\{0,1\}^n$ and $|\mathcal{A}|=O(\mathsf{poly}(n))$, the circuit complexity of the optimal value function will not fall below the optimal policy under the $\mathsf{TC}^0$ reduction.
>
>
> > However, our representation complexity hierarchy is not valid for all MDPs.
> For instance, in MDPs characterized by complex transition kernels and zero reward functions, the model's complexity surpasses that of the optimal policy and value function (*this aligns with the "counterexample" noted by the reviewer*). However, these additional MDP classes may not be typical in practice and could be considered pathological examples from a theoretical standpoint. We leave the fully theoretical characterizing of representation hierarchy between model-based RL, policy-based RL, and value-based RL as an open problem.  For instance, it could be valuable to develop a methodology for classifying MDPs into groups and assigning a complexity ranking to each group within our representation framework.
>
>
>
> **Q2:** This also makes the empirical section a bit more difficult: Why should we assume that the Mujoco locomotion environments fall into the category of problems described in the paper and not into the alternative? Empirically this seems to be true. Is there some reason to believe that "realistic" MDPs will exhibit this complexity relationship more, or is this an artifact of the choice of environments?
>
> **A2:** Thank you for your question. It's difficult to argue that Mujoco environments fall into the category of constructed MDPs for theoretical understanding. However, constructed MDPs and Mujoco environments (or more general real-world applications) share similar features: they encode complex decision-making problems into relatively simple models (reward and transition). Our theoretical framework aims to characterize a broad range of real-world problems by incorporating $\mathsf{P}$ or $\mathsf{NP}$ problems into the transition. This way of study aligns with the theoretical understanding of deep learning, where researchers typically demonstrate or explain phenomena (such as benign overfitting) in ideal theoretical models (such as linear regression or two-layer neural networks). Moreover, we want to emphasize that our work uses a completely new perspective --- the expressive power of neural networks --- to study RL problems, which is highly related to our experiments and can be considered an important step in bridging the gap between theory and practice. Finally, for the reasons stated above, we believe the demonstrated representation hierarchy, although not universal, characterizes a wide range of RL problems and is definitely not an artifact of the choice of environments.
>
> ---
>
> We sincerely hope the reviewer will reconsider their rating of our work, which is the first comprehensive study of representation complexity. We are also open to further discussion if the reviewer has any concerns about the correctness of our proof or the contribution of our research.

---

> > ### Comment · Reviewer_YjQh · 2024-08-11
> > **Answer**
> >
> > Dear authors, thanks for the clarifications. I acknowledge that I missed the discussion in appendix C, I feel like this is a very important point that might deserve some more prominent space in the main paper.
> >
> > I still don't fully agree with the argument: the benchmarks are investigated here are all designed to have little to no task irrelevant features such as distractions. Generalizing from these to RL problems in general seems like it can mislead the community. The results also violate a lot of commonly held intuition: in many cases predicting the exact consequences of your actions is very hard, i.e. how many blades of grass are crushed by a step on the lawn, but from the perspective of any task, this is irrelevant. I wish I could provide a nice reference here, but I can't find one at the moment.
> >
> > I don't have a problem at all with your formal statements and I do applaud the novel and insightful technique. My whole problem is with the nuance of the framing, which I do think is very important to get right to frame your (very interesting results) in the right way.
> >
> > I am willing to increase my score provided you can give the nuance of this discussion some space in the main paper. Since I want to be an optimistic and constructive reviewer, I have updated my score to recommend acceptance, I hope you follow my concerns.

---

> > > ### Author Response · Authors · 2024-08-13
> > >
> > > Thank you for taking the time to read our response and update the score. We appreciate your feedback and will certainly follow your suggestions to polish our paper.

---

### Official Review · Reviewer_4u4n · 2024-07-13

**Soundness:** 3
**Presentation:** 3
**Contribution:** 3
**Rating:** 5
**Confidence:** 2

**Summary:**

This paper studies three RL paradigms: model-based RL, policy-based RL, and value-based RL from the perspective of representation complexity. The authors demonstrate that representing the model emerges as the easiest task, followed by the optimal policy, with the optimal value exhibiting the highest representation complexity.

**Strengths:**

- The paper is well written. The problem studied in this paper is well-motivated and very interesting.
- Analyzing model-based RL, policy-based RL, and value-based RL from the perspective of representation complexity, transitioning from simple scenarios to broader ones, is impressive.

**Weaknesses:**

- From the perspective of MLP, using simple 2 or 3-layer MLPs to calculate approximation error to validate conclusions provides limited insights for modern deep RL.
- Although it is a theory paper, I would have liked to see more experiments designed to validate the conclusions.

**Questions:**

- Intuitively, could you explain in detail the fundamental reasons for the different representation complexities of the optimal policy and optimal value function under different settings in Sec. 3 and Sec. 4?
- Maybe you should describe more examples to intuitively understand the representation complexity of the model, optimal policy, and optimal value function.

**Limitations:**

The paper is mainly theoretical, and no algorithm is implemented. There is no specific potential negative societal impact of this work.

---

> ### Author Rebuttal · Authors · 2024-08-07
>
> Thank you for taking the time to review our paper. We appreciate your feedback and will address each of your concerns individually.
>
> ---
>
> **W1:** From the perspective of MLP, using simple 2 or 3-layer MLPs to calculate approximation error to validate conclusions provides limited insights for modern deep RL.
>
> **Response:** We appreciate the reviewer's point regarding the complexity of MLPs in modern deep RL. However, our experiments are conducted in Mujoco environments where the approximation challenges are relatively modest. In these settings, MLPs with 3 layers and 256 hidden units each have been shown to perform near the state-of-the-art (SOTA) when utilizing algorithms such as TD3 (and SAC...). This indicates that the complexity of the function approximators required for these environments is not high. Consequently, using simple 2 or 3-layer MLPs as our testbed is a reasonable choice for the scope of our study. Moreover, as Mujoco environments are classic and extensively utilized benchmarks within the deep RL community, insights gained from these experiments hold significant relevance.
>
> **W2:** Although it is a theory paper, I would have liked to see more experiments designed to validate the conclusions.
>
> **Response:** We recognize the value that additional experiments would bring, particularly in complex environments (such as robotics), to reinforce our conclusions. Unfortunately, due to constraints in resources, extensive experimentation in diverse real-world scenarios was not feasible within the scope of this study. However, we selected Mujoco environments for our experiments because they are classic and widely recognized benchmarks in the deep reinforcement learning community. The experiments are conducted in 4 environments and repeated with 5 random seeds, and the results are therefore fairly sufficient to support our theoretical results and contribute to the general understanding of our theoretical claims. If the reviewer has more concrete suggestions to enhance the experiments, we are happy to incorporate them.
>
> **Q1 & Q2:** (1) Intuitively, could you explain in detail the fundamental reasons for the different representation complexities of the optimal policy and optimal value function under different settings in Sec. 3 and Sec. 4? (2) Maybe you should describe more examples to intuitively understand the representation complexity of the model, optimal policy, and optimal value function.
>
> **Response:** Thank you for your question. The fundamental difference in representation complexity for optimal policies and value functions in Sec. 3 and Sec. 4 stems from the nature of the decision-making process in various reinforcement learning (RL) scenarios.
>
> In Sec. 3, we consider environments with long-term dependencies where the optimal policy needs to encapsulate complex strategies due to the high branching factor of possible future state trajectories. Here, the representation complexity of both the policy and the value function is high since they must incorporate information about the consequences of actions over many time steps. This is akin to games like chess, where the policy must be sophisticated enough to navigate a vast tree of possible moves.
>
> In contrast, Sec. 4 addresses environments with shorter planning horizons. The optimal policy in such settings can be less complex because it's more focused on immediate states which is sufficient to make a good decision. However, the optimal value function may still require a complex representation to accurately predict long-term returns from any given state due to the potential variability in future rewards. An example provided is a robotic gripping task where the immediate action (gripping with the correct force) is straightforward, but the long-term implications (successfully gripping various objects without dropping) add complexity to the value function.
>
> In the [Mujoco environment](https://www.gymlibrary.dev/environments/mujoco/index.html)(e.g. [Ant-v4](https://www.gymlibrary.dev/environments/mujoco/ant/)), the transition kernels and reward functions of these environments are some simple functions containing several rules. However, as for the optimal policy and optimal value function, they are so complex that need a complex neural network to approximate and a well-designed RL algorithm to learn.
>
> In summary, the complexity of the optimal policy and value function representations is intrinsically linked to the depth of foresight required for decision-making in a given RL scenario. We will add more analysis in our paper to illustrate this distinction and its implications for the design of RL algorithms.
>
> ---
>
> Please let us know if you have any further questions. If your concern is addressed, we would appreciate it if you would reconsider your score in light of our clarification.

---

> > ### Author Response · Authors · 2024-08-13
> >
> > Dear reviewer,
> >
> > Thank you again for reviewing our paper. As the discussion period ends soon, we'd like to know if our response has adequately addressed your concerns. If not, we're happy to provide further clarification. We greatly appreciate your time and support.
> >
> > Best,
> > Authors

---

> > ### Comment · Reviewer_4u4n · 2024-08-13
> >
> > Thank you for your detailed response to my review. The authors' rebuttal has addressed some of my concerns. My intuition remains in favor of acceptance, though my limited knowledge in the area of representation complexity.

---

### Official Review · Reviewer_UVj9 · 2024-07-13

**Soundness:** 3
**Presentation:** 3
**Contribution:** 2
**Rating:** 5
**Confidence:** 3

**Summary:**

This paper delves into understanding the inherent representation complexities associated with three different RL categories: model-based RL, policy-based RL, and value-based RL. By studying computational complexity theory and neural networks, MLP, the paper posits a hierarchy in which representing the underlying model is the simplest, followed by the optimal policy, and finally, the optimal value function, which is the most complex to represent. Theoretical analyses and empirical results support these claims.

**Strengths:**

- The paper gives a new understanding and improves RL algorithms by examining RL paradigms through the lens of representation complexity.
- The paper includes deep RL experiments that align with the theoretical findings, offering practical evidence of the proposed hierarchy.
-The paper bridges theoretical insights with deep RL by discussing the expressiveness of MLPs.

**Weaknesses:**

- While the theoretical insights are significant, the paper does not extensively explore their direct implications for real-world RL applications.
- Previous experiments show model-based RL is more sample-efficient than model-free RL, aligning with this paper's finding that representing the dynamic model is easier. However, as training progresses, model-free methods often outperform model-based ones. Could the theories in the paper explain this? While the policy is more challenging to represent initially, it allows model-free methods to optimize more efficiently once learned.
- Could the proposed hierarchy of representation complexity be applied to other types of neural network architectures beyond MLPs, such as transformers?

**Questions:**

Please answer the points mentioned in the weaknesses.

**Limitations:**

The paper does not discuss their limitations.

---

> ### Author Rebuttal · Authors · 2024-08-07
>
> Thank you for taking the time to review our paper. We appreciate your feedback and will address each of your concerns individually.
>
> **Q1:** While the theoretical insights are significant, the paper does not extensively explore their direct implications for real-world RL applications.
>
> **A1:** Thank you for recognizing the theoretical insights of our work. In fact, our research empirically demonstrates a consistent representation complexity hierarchy, accompanied by a theoretical understanding from the perspective of deep neural networks (a completely new perspective in RL theory!). Our work also has implications for explaining why model-based RL is more sample-efficient. We view our work as the beginning of a comprehensive understanding of representation complexity in RL; therefore, further direct implications for RL algorithms in real-world applications are left for future work.
>
>
> **Q2:** Previous experiments show model-based RL is more sample-efficient than model-free RL, aligning with this paper's finding that representing the dynamic model is easier. However, as training progresses, model-free methods often outperform model-based ones. Could the theories in the paper explain this? While the policy is more challenging to represent initially, it allows model-free methods to optimize more efficiently once learned.
>
> **A2:** Our theoretical analysis focuses on the **representation complexity** of RL paradigms, explaining the initial sample efficiency of model-based methods. We recognize that as training advances, the direct policy representation in model-free methods may lead to more efficient optimization, potentially outperforming model-based approaches.  This difference primarily stems from the distinct **optimization properties** of various reinforcement learning algorithms.
>
>
> We emphasize that **representation complexity and optimization properties are two parallel and equally important aspects**. Our work fills a gap in understanding representation complexity in RL theory. We fully agree with the reviewer that integrating optimization efficiency with representation complexity to comprehensively understand the performance differences between various RL algorithms is a promising (and challenging) direction for future research. We appreciate the reviewer highlighting this important area of investigation.
>
>
> **Q3:** Could the proposed hierarchy of representation complexity be applied to other types of neural network architectures beyond MLPs, such as transformers?
>
> **A3:** Our hierarchy of representation complexity **can be applied to the transformer architecture**. Please see the **general response** for the details.
>
> ---
> Please let us know if you have any further questions. If your concern is addressed, we would appreciate it if you would reconsider your score in light of our clarification.

---

> > ### Comment · Reviewer_UVj9 · 2024-08-11
> >
> > Thanks for the rebuttal! My concerns have been addressed, so I decide to keep my positive rating.

---

> > > ### Author Response · Authors · 2024-08-13
> > >
> > > We are happy to hear that we've addressed your concerns. Thank you for your valuable time reviewing our paper.

---

### Official Review · Reviewer_eFNz · 2024-07-14

**Soundness:** 3
**Presentation:** 3
**Contribution:** 3
**Rating:** 7
**Confidence:** 2

**Summary:**

This paper delves into the representation complexity in different RL paradigms. It focuses on the function class needed to represent the underlying model, optimal policy, or optimal value function.

**Strengths:**

1. The study uses time complexity and circuit complexity to theoretically analyze the representation complexity among RL paradigms. It introduces new classes of MDPs (3-SAT MDPs, NP MDPs, CVP MDPs, and P-MDPs) to showcase the differences in complexity.
2. The paper finds that models and policies can often be effectively represented by MLPs, while optimal value functions face limitations, providing insights that may inform future research.

**Weaknesses:**

See questions.

**Questions:**

I apologize for my limited familiarity with the representation complexity area. I have read the paper thoroughly and examined the theorems. I appreciate your work, but I feel I cannot make a solid judgment. So if my questions are off the mark, please correct me.

1. Are the findings regarding the representation complexity hierarchy consistent across a wide range of task settings, or do they vary significantly with different types of tasks?
2. Can you provide case studies or real-world examples for understanding the representation complexity hierarchy that has led to improved sample efficiency in practical applications? I understand you have provided some explanations in the paper. If this question adds extra burden, feel free to disregard it.

**Limitations:**

The authors do not have an explicit limitations section or paragraph.

---

> ### Author Rebuttal · Authors · 2024-08-07
>
> We sincerely thank the Reviewer for the positive feedback and appreciation for our work.
>
> **Q1:** Do the findings regarding the representation complexity hierarchy hold consistently across different task settings, or do they vary significantly with various types of tasks?
>
> **Response:** Yes, our findings on the representation complexity hierarchy demonstrate consistent behavior across a diverse array of task settings. (i) Theoretically, we can encode *any* $\mathsf{NP}$ or $\mathsf{P}$ problem into the construction of MDPs and establish the desired representation complexity hierarchy, characterizing a wide range of problems. (ii) Empirically, we have conducted experiments across various simulated environments, each designed to test different aspects of task complexity. These environments range from simple binary classification tasks to more complex structured prediction problems. In all these settings, the hierarchy we have identified remained robust and consistent. Thank you for raising this important question and we will further emphasize this in our revision.
>
>
>
> **Q2:** Can you provide case studies or real-world examples for understanding the representation complexity hierarchy that has led to improved sample efficiency in practical applications?
>
> **Response:** Thank you for your question. Our research findings indicate that the model typically benefits from the lowest representation complexity, which may explain why model-based reinforcement learning generally achieves better sample efficiency in real-world applications (see Appendix C.1 for more details). Moreover, we'd like to emphasize that our work primarily focuses on providing the first comprehensive theoretical understanding of representation complexity across various reinforcement learning paradigms. We hope this understanding will inspire the following empirical works and lead to practical advancements in sample efficiency.

---

> > ### Comment · Reviewer_eFNz · 2024-08-08
> > **Thank the authors for the rebuttal!**
> >
> > Thank you for the detailed rebuttal! My concerns have been resolved.
> > Apologize again for my limited knowledge in your area.
> > Wish you good luck with this paper!

---

> > > ### Author Response · Authors · 2024-08-09
> > >
> > > We are pleased that our response has addressed your concerns. Thank you for taking the time to review our paper.

---

### Author Rebuttal · Authors · 2024-08-07

We appreciate all the reviewers for reviewing our paper. We have provided comprehensive responses separately and demonstrate in our general response that our findings can be extended to deep reinforcement learning with **transformer** architectures.

Here are some informal theorems and proof sketches. We will incorporate these theorems in the next version of this paper.

> **Theorem:** Assuming that $\mathsf{TC}^0\neq\mathsf{NP}$, the optimal policy $\pi_1^*$ and optimal value function $Q_1^*$ of $n$-dimensional 3-SAT MDP and $\mathsf{NP}$ MDP defined with respect to an $\mathsf{NP}$-complete language $\mathcal{L}$ cannot be represented by a Transformer with constant layers, polynomial hidden dimension (in $n$), and ReLU as the activation function.

> **Theorem:** Assuming that $\mathsf{TC}^0\neq\mathsf{P}$, the optimal value function $Q_1^*$ of $n$-dimensional CVP MDP and $\mathsf{P}$ MDP defıned with respect to a $\mathsf{P}$-complete language $\mathcal{L}$ cannot be represented by a Transformer with constant layers, polynomial hidden dimension (in $n$), and ReLU as the activation function.

> **Proof:**  According to Lemma I.7 in our paper, and the previous work[1], a Transformer with logarithmic precision, a fixed number of layers, and a polynomial hidden dimension can be simulated by a  $\mathsf{L}$-uniform $\mathsf{TC}^0$ circuit. On the other hand, the computation of the optimal policy and optimal value function for the 3-SAT MDP and NP MDP is NP-complete, and the computation of the optimal value function for CVP MDP and P MDP is P-complete. Therefore, the theorem holds under the assumption of $\mathsf{TC}^0\neq\mathsf{NP}$ and $\mathsf{TC}^0\neq\mathsf{P}$.

> **Theorem:** The reward function $r$ and transition kernel $\mathcal{P}$ of $n$-dimensional 3-SAT MDP and NP MDP can be represented by a Transformer with constant layers, polynomial hidden dimension (in $n)$, and ReLU as the activation function.

> **Theorem:** The reward function $r$, transition kernel $\mathcal{P}$, and optimal policy $\pi^*$ of $n$-dimensional CVP MDP and P MDP can be represented by a Transformer with constant layers, polynomial hidden dimension (in $n)$, and ReLU as the activation function.

> **Proof Sketch:** It is important to note that the MLP is a submodule of the Transformer. According to Theorems 5.2 and 5.4, an MLP with constant layers, polynomial hidden dimension (in $n)$, and ReLU activation can represent these functions. Given an input sequence of states, the transformer can just use the MLP module to calculate the corresponding functions.

[1] William Merrill and Ashish Sabharwal. The parallelism tradeoff: Limitations of log-precision transformers. Transactions of the Association for Computational Linguistics.

---

### Decision · Program_Chairs · 2024-09-25

**Decision:**

Accept (poster)

**Comment:**

The paper analyzes a hierarchy of representation complexity in RL. The reviewers found the paper difficult to read and sometimes needed the help of the authors in identifying key information in the appendices. The authors and reviewers have engaged in a number of rounds of rebuttal, and the reviewers have proposed to accept the paper conditioned that the authors address the nuances discussed in appendix C with a more prominent space in the main paper. Conditioned on addressing this issue, we recommend acceptance as a poster.
Thank you